# SARS-CoV-2 humoral and cellular immunity following different combinations of vaccination and breakthrough infection

Jernej Pušnik[1,2], Werner O. Monzon-Posadas[1,2,3], Jasmin Zorn[1,2], Kathrin Peters [1,2], Maximilian Baum[1,2], Hannah Proksch [1,2], Celina Beta Schlüter[1,2], Galit Alter [4], Tanja Menting[3] & Hendrik Streeck [1,2] ✉

The elicited anti-SARS-CoV-2 immunity is becoming increasingly complex with individuals receiving a different number of vaccine doses paired with or without recovery from breakthrough infections with different variants. Here we analyze the immunity of individuals that initially received two doses of mRNA vaccine and either received a booster vaccination, recovered from a breakthrough infection, or both. Our data suggest that two vaccine doses and delta breakthrough infection or three vaccine doses and optionally omicron or delta infection provide better B cell immunity than the initial two doses of mRNA vaccine with or without alpha breakthrough infection. A particularly potent B cell response against the currently circulating omicron variant (B. 1.1.529) was observed for thrice vaccinated individuals with omicron breakthrough infection; a 46-fold increase in plasma neutralization compared to two vaccine doses ($p < 0.0001$). The T cell response after two vaccine doses is not significantly influenced by additional antigen exposures. Of note, individuals with hybrid immunity show better correlated adaptive immune responses compared to those only vaccinated. Taken together, our data provide a detailed insight into SARS-CoV-2 immunity following different antigen exposure scenarios.

The worldwide vaccination campaign against SARS-CoV-2 infections demonstrated outstanding results in cutting down the severity of infections[1–3]. The first approved vaccines were based on the novel mRNA technology and triggered a robust immune response resulting in high protection efficiency[4–7]. This was reflected in the low frequency of breakthrough infections among the vaccinated, which were mostly caused by the alpha variant (Pango lineage B.1.1.7), and resulted in mild disease[8,9]. However, similar to infection-induced immunity, the immune response to SARS-CoV-2 vaccines rapidly declines below the level required for protection from infection[10,11]. By the time the first vaccines were rolled out, the SARS-CoV-2 pandemic was already established in most of the populated regions on earth[12]. With 100,000 new infections per week globally[12], the virus had plenty of space to mutate and adapt in a way that would overcome the immune protection of the vaccinated population. The waning immunity and the emergence of new viral variants like delta (Pango lineage B.1.617.2) fuelled the increasing frequency of breakthrough infections[13,14]. Although severe Covid-19 cases were less frequent in previously vaccinated than unvaccinated individuals it became clear that booster vaccinations will be needed to maintain protection from severe SARS-CoV-2 infection[15,16]. In developed countries like Germany, the majority of the vaccinated individuals received a third vaccine dose significantly

[1]Institute of Virology, University Hospital Bonn, Bonn 53127, Germany. [2]German Center for Infection Research (DZIF), partner site Bonn-Cologne, Braunschweig 38124, Germany. [3]Occupational Medicine Department, University Hospital Bonn, Bonn 53127, Germany. [4]Ragon Institute of MGH, MIT, and Harvard, Massachusetts General Hospital, Boston, MA 02139-3583, USA. ✉e-mail: Hendrik.Streeck@ukbonn.de

improving the immune response[17,18]. Recently, a fourth vaccine dose was recommended for individuals at high risk for severe disease. However, in particular, against the currently circulating omicron variant (Pango lineage B. 1.1.529) the vaccine-induced immunity offers little protection from infection as seen in high rates of breakthrough infections[12]. Protection from severe diseases nevertheless remains high among vaccinated individuals[19].

SARS-CoV-2 infection of vaccinated individuals rarely develops a severe disease course, and most of the infections resolve without life-threatening consequences[14]. Importantly, these breakthrough infections further strengthen the immunity established by vaccination. Studies have shown that vaccinated individuals that also recovered from a SARS-CoV-2 infection have comparable neutralizing antibody titers to those that received three vaccine doses, and are therefore better protected from the severe disease than twice-vaccinated individuals[20–23]. Furthermore, breakthrough infections compensate for the waning of immunity established by previous infection or vaccination[20]. However, there is heterogeneity in breakthrough infections amongst vaccinated individuals in terms of the total number of antigen exposures (defined as an immune response to infection or vaccination throughout the manuscript) and the variant that causes the infection. We postulated that individuals that initially received two doses of mRNA vaccine and later became infected with different variants and/or received a third vaccine dose have distinct immunity profiles depending on the antigen exposure. In this observational study, we compared the antibody levels, B cell, and T cell responses amongst the individuals belonging to seven different groups based on their antigen exposure history: 2-times vaccinated (2xVacc), 2-times vaccinated followed by alpha breakthrough infection (2xVacc+α), 2-times vaccinated followed by delta breakthrough (2xVacc+δ), 3-times vaccinated (3xVacc), 3-times vaccinated followed by omicron infection (3xVacc+o), 3-times vaccinated where the third dose was preceded by an alpha infection (3xVacc+α), and 3-times vaccinated where the third dose was preceded by a delta infection (3xVacc+δ).

## Results

### Not only the number but also the type of antigen exposure is important for potent humoral immunity against the SARS-CoV-2

Antibodies are the best-defined correlate of protection against the SARS-CoV-2 infection[24], therefore, we first investigated levels of neutralizing antibodies among individuals with different antigen exposure histories. Based on the antigen exposure status we defined the following groups: 2-times vaccinated (2xVacc, $n = 54$), 2-times vaccinated followed by alpha breakthrough infection (2xVacc+α, $n = 7$), 2-times vaccinated followed by delta breakthrough (2xVacc+δ, $n = 13$), 3-times vaccinated (3xVacc, $n = 23$), 3-times vaccinated followed by omicron infection (3xVacc+o, $n = 10$), 3-times vaccinated where the third dose was preceded by an alpha infection (3xVacc+α, $n = 7$), and 3-times vaccinated where the third dose was preceded by a delta infection (3xVacc+δ, $n = 7$) (Fig. 1a). The detailed information on the antigen exposure and sampling time points along with the demographic information is provided in the supplement (Supplemental Fig. 1 and Supplemental Table 1).

Determination of plasma IgG titer against the S1 subunit of the SARS-CoV-2 spike protein was carried out by the in-house ELISA calibrated to the international WHO standard (NIBSC reference number: 20/136). The data revealed that 2xVacc+δ, 3xVacc, 3xVacc+o, and 3xVacc+δ groups had significantly higher S1-specific IgG titers compared to the 2xVacc group and the groups with an alpha breakthrough infection; 2xVacc+α, 3xVacc+α (Fig. 1b). The largest difference (5.2-fold, $p < 0.0001$) was observed between the 2xVacc+δ and 2xVacc groups. To investigate whether the same pattern can be observed for only neutralizing antibodies we performed plaque reduction assays using live unmanipulated SARS-CoV-2 isolates: wild-type (Pango lineage A), delta, and omicron variants. We observed similar differences

between the groups as for the S1-binding IgG measured by ELISA for all variants. Of note, the 2xVacc+δ group showed particularly high neutralization potency against the wild-type (6.5-fold compared to 2xVacc, $p < 0.0001$) and delta variants (7.8-fold compared to 2xVacc, $p < 0.0001$), while the 3xVacc+o group was the most efficient at neutralizing the omicron variant (46-fold compared to 2xVacc, $p < 0.0001$) (Fig. 1c). Comparing the three SARS-CoV-2 variants and their susceptibility to neutralization we observed that the omicron variant was significantly more resistant to neutralization than delta and wild-type. The most profound reduction in the neutralization capacity against the omicron variant was observed for the 2xVacc and 2xVacc+α groups (16-fold and 20-fold respectively when compared to the wild-type). The only exception was the 3xVacc+o group that equally neutralized all three variants. Importantly, delta showed significantly higher resistance to neutralization than the wild-type exclusively in the case of the groups without breakthrough infections; 2xVacc, 3xVacc (Fig. 1d). To assess the relationship between the S1-specific IgG levels and neutralization capacity against the wild-type virus we correlated the two parameters for each of the antigen exposure groups. The level of S1-binding antibodies correlated with neutralization regardless of the antigen exposure history. The strongest correlations were observed in case of the 2xVacc+α ($r = 0.89$, $p = 0.012$), 3xVacc+α ($r = 0.93$, $p = 0.0067$), 2xVacc+δ ($r = 0.93$, $p < 0.0001$), and 3xVacc+δ ($r = 0.82$, $p = 0.034$) groups (Fig. 1e). Moreover, we correlated the S1-specific IgG titers with neutralization of the delta variant. The groups with the strongest correlations were in this case 2xVacc+δ ($r = 0.93$, $p < 0.0001$), 3xVacc+δ ($r = 0.86$, $p = 0.024$), and 2xVacc+α ($r = 0.89$, $p = 0.012$) (Fig. 1f). Correlations with neutralization of the omicron variant were notably weaker compared to the other two variants and significant only for the 3 out of 7 groups; 2xVacc, 2xVacc+δ, and 3xVacc+o. Relatively strong associations were observed only for the 2xVacc+δ ($r = 0.81$, $p = 0.0014$) and 3xVacc+o ($r = 0.81$, $p = 0.0082$) groups (Fig. 1f). Furthermore, we assessed the presence of antibodies specific for the nucleocapsid (N) protein of the SARS-CoV-2 using a commercial Roche Cobass assay. Complying with findings for the S1-specific IgG levels we observed that 77% of the delta, and 100% of omicron, but only 43% of the alpha breakthrough infections led to seroconversion.

Taken together our findings demonstrate that two vaccine doses and delta breakthrough infection or three vaccine doses and optionally omicron or delta infection provide significantly better humoral immunity to SARS-CoV-2 infection compared to the baseline antigen exposure with 2 doses of mRNA vaccine. A particularly strong humoral response against different SARS-CoV-2 variants was observed among twice-vaccinated individuals that got infected with the delta variant and thrice-vaccinated individuals that recovered from omicron breakthrough infection.

### Salivary S1-specific antibody levels are highly dependent on the antigen exposure scenario

SARS-CoV-2 initially replicates in the upper respiratory tract where it is exposed to antibodies present in mucosa and saliva[25]. Therefore, the SARS-CoV-2-specific antibody titer in saliva might be a better correlate of protection from infection than the level of antibodies in plasma[26,27]. To determine the titer of IgG and IgA specific for the S1 subunit of the SARS-CoV-2 spike protein we developed an ELISA detecting comparatively low amounts of antibodies present in saliva. We demonstrated that similar to S1-specific plasma IgG levels, 2xVacc+δ, 3xVacc, 3xVacc+o, and 3xVacc+δ groups had significantly higher IgG levels compared to the baseline 2xVacc group, while the groups with an alpha breakthrough infection; 2xVacc+α, 3xVacc+α had relatively few S-specific IgG in saliva. Particularly high S1-specific IgG levels were observed in the 3xVacc+o group which had a 13-fold higher titer than the baseline 2xVacc group ($p < 0.0001$) (Fig. 2a). In contrast to IgG, salivary S1-specific IgA levels were only significantly increased in the

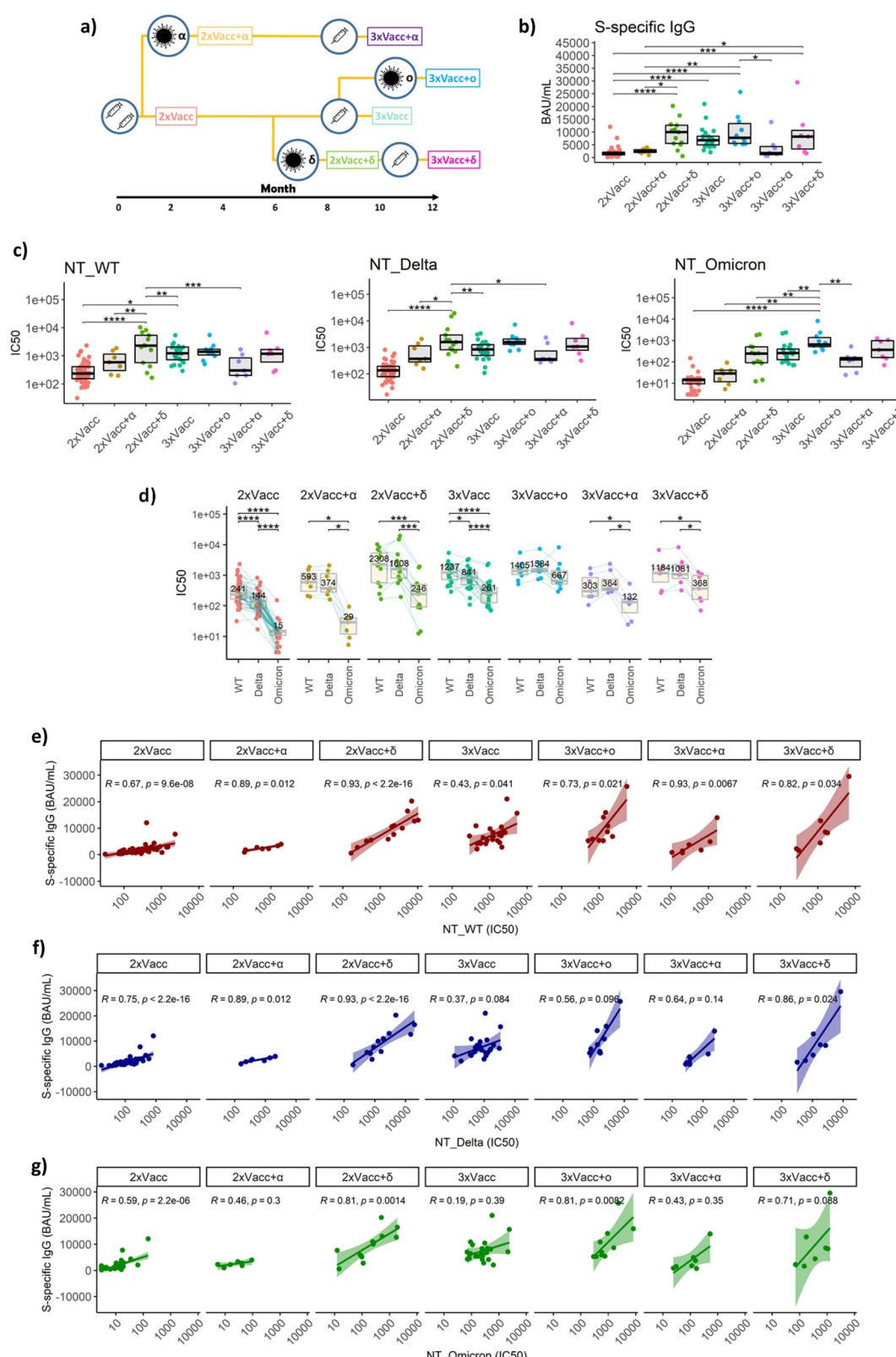

case of 2xVacc+δ and 3xVacc+o groups (Fig. 2b). Since we used the same monoclonal antibody with variable Fc region as a calibrator for the IgG and IgA ELISAs we were able to compare the relative binding strengths of the two antibody isotypes. The highest cumulative binding strength of S1-specific IgG and IgA was observed for the 3xVacc+o group (a 9.5-fold increase compared to 2xVacc; $p < 0.0001$) followed by 2xVacc+δ, 3xVacc, and 3xVacc+δ. 2xVacc and 2xVacc+α groups

were significantly lower than most of the other groups (Fig. 2c). Regardless of the antigen exposure group, IgG showed a stronger binding capacity than IgA. The highest proportion of IgA binding was observed for the 2xVacc group and the lowest for the 3xVacc group (Fig. 2d). We next investigated whether S1-specific IgG levels in saliva reflect those measured in plasma. In contrast to plasma, the only significant correlations in saliva were observed for the 2xVacc ($r = 0.28$,

**Fig. 1 | Plasma antibody response after different combinations of vaccination and breakthrough infection. a** Schematic chronological representation of antigen exposures defining the seven groups compared in this study. Detailed information about the antigen exposure and sampling time points is provided as supplemental material. **b** S1-specific IgG levels in international units measured for the seven groups with different antigen exposure histories. Exact *p* values in sequential order from the upmost bracket: *\**p* = 0.028, \*\*\**p* = 0.00021, *\**p* = 0.032, \*\**p* < 0.0038, \*\*\*\**p* < 0.0001, \*\*\*\**p* < 0.0001, *\**p* = 0.011, \*\*\*\**p* < 0.0001. **c** Plasma neutralization capacity against wild-type, delta, and omicron variants. Exact *p* values in sequential order from the upmost bracket of the left graph: \*\*\**p* = 0.00071, \*\**p* = 0.0027, *\**p* = 0.021, \*\**p* = 0.0023, \*\*\*\**p* < 0.0001, *\**p* = 0.04, \*\**p* = 0.0034, *\**p* = 0.04, \*\*\*\**p* < 0.0001, \*\**p* = 0.0035, \*\**p* = 0.0018, \*\**p* = 0.0081, \*\**p* = 0.0011, \*\*\*\**p* < 0.0001. **d** Comparison of neutralization susceptibility of SARS-CoV-2 variants for each of the analyzed groups. The IC50 medians are given numerically for each boxplot. Exact p values in sequential order from the upmost bracket of the leftmost group:

\*\*\*\**p* < 0.0001, \*\*\*\**p* < 0.0001, \*\*\*\**p* < 0.0001, *\**p* = 0.047, *\**p* = 0.047, \*\*\**p* = 0.00073, \*\*\**p* = 0.00073, \*\*\*\**p* < 0.0001, *\**p* = 0.023, \*\*\*\**p* < 0.0001, *\**p* = 0.047, *\**p* = 0.047, *\**p* = 0.047, *\**p* = 0.047. In panels **b–d** the data is displayed as boxplots, indicating the first quartile, median, and third quartile, with individual data points. Correlation between plasma levels of S1-specific IgG and plasma neutralization capacity for **e** wild-type, **f** delta, **g** omicron variants. The 95% confidence intervals around the line of best fit are displayed as shading. The *r* and *p* values are given for each line. The following numbers of biologically independent samples were included in each group for all the graphs in this figure: 2xVacc, *n* = 54; 2xVacc+α, *n* = 7; 2xVacc+δ, *n* = 13; 3xVacc, *n* = 23; 3xVacc+o, *n* = 10; 3xVacc+α, *n* = 7; 3xVacc+δ, *n* = 7. Differences between the groups were assessed using the Mann–Whitney test or Wilcoxon test for matched data. The strength of correlations was assessed by Spearman's correlation test. Correction for multiple testing was performed using Holm's method, all statistical tests were two-sided. Source data are provided as a Source data file.

---

*p* = 0.044), 2xVacc+α (*r* = 0.93, *p* = 0.0067), and 2xVacc+δ (*r* = 0.75, *p* = 0.012) groups (Fig. 2e).

Collectively, we have shown that the individuals that received 3 vaccine doses and afterward acquired an omicron infection show the highest S1-specific antibody titer in saliva. Interestingly, IgG represented the majority of S1-binding antibodies in saliva and its levels correlated with S1-specific plasma IgG only in cases of particular antigen exposure combinations.

## The frequency of S1-specific memory B cells is influenced by the antigen exposure history

When assessing the quality of SARS-CoV-2 immunity it is important to not only consider the humoral but also cellular components that are particularly important for long-term protection from the disease[28].

We, therefore, measured the frequency of S1-specific memory B cells in the peripheral blood of individuals with different antigen exposure histories utilizing multiparameter flow-cytometry (Fig. 3a). Detailed gating strategy for identification of S1-specific memory B cells is provided in Supplemental Fig. 2. Our data demonstrate increased frequencies of IgG+ S1-specific memory B cells for 2xVacc+δ, 3xVacc, and 3xVacc+o groups when compared to the 2xVacc and 2xVacc+α groups. In the case of the 3xVacc+δ and 3xVacc+α groups, the frequencies of these cells were relatively low and comparable to the baseline 2xVacc group (Fig. 3b). No statistically significant differences between the antigen exposure groups were observed for the IgA+ and IgM+ S1-specific memory B cells (Fig. 3c, d, respectively). When considering the frequency of the total S1-specific memory B cells, regardless of the B cell receptor (BCR) isotype, the 3xVacc+o group had the highest frequency of these cells (a 3.5-fold increase compared to the 2xVacc group; *p* < 0.05) followed by 2xVacc+δ and 3xVacc groups. The frequencies observed within 3xVacc+δ, 3xVacc+α, and 2xVacc+α groups were comparable to the 2xVacc group (Fig. 3e). We next investigated the relative proportions of S1-specific memory B cells with different BCR isotypes. We observed that IgG+ S1-specific memory B cells were the most frequent for all groups compared (*p* < 0.001). 2xVacc and 2xVacc+α groups had significantly increased levels of non-IgG+ S1-specific memory B cells compared to 2xVacc+δ and 3xVacc+δ groups (*p* < 0.05 for all comparisons) (Fig. 3f).

Taken together our findings demonstrate increased frequencies of S1-specific memory B cells among individuals that either received 3 doses of mRNA vaccine, 3 doses of mRNA vaccine plus omicron infection, or 2 doses of vaccine and delta infection. IgG was the most prevalent BCR isotype of S1-specific memory B cells regardless of the antigen exposure history.

## Different antigen exposure combinations do not significantly affect SARS-CoV-2-specific T cells

T cells contribute to the defense against viral infections by coordinating the production of antibodies and killing the infected cells. It has

been previously shown that T cells specific for SARS-CoV-2 successfully limit the infection and positively correlate with protection from severe disease[29,30].

Given their importance we next measured the frequencies of CD4+ and CD8+ T cells specific for the spike (S) protein of the SARS-CoV-2 in the peripheral blood of individuals with different antigen exposure histories. Antigen-specific cells were detected by peptide stimulation and subsequent detection of cytokine expression by multiparameter flow cytometry. Four major functions of the T cells were monitored; cytotoxicity (CD107a and IFNγ expression), IFNγ expression, IL-2 expression, and TNFα expression (Fig. 4a). Detailed gating strategy for identification of antigen-specific T cells is provided as Supplemental Fig. 3. Interestingly, we did not observe any significant differences between the antigen exposure groups for the S-specific CD4+ T cells regardless of their function. Nevertheless, a similar trend was observed in the cases of cytotoxic, IFNγ-expressing and IL-2-expressing CD4+ T cells as for the S1-specific plasma IgG; 2xVacc+δ, 3xVacc, 3xVacc+o, and 3xVacc+δ were higher than the baseline 2xVacc group and the groups with an alpha breakthrough infection; 2xVacc+α, 3xVacc+α (Fig. 4b). Also, the frequencies of S-specific CD8+ T cells were not significantly different between the antigen exposure groups. However, there was a trend suggesting that the 3xVacc+o group had the highest median frequency of those cells for all groups (Fig. 4c). We have previously demonstrated that unvaccinated individuals infected with SARS-CoV-2 develop high frequencies of the nucleocapsid-specific T cells[31,32]. To check whether this is also true for breakthrough infections we stimulated the peripheral blood T cells with peptides spanning the entire sequence of the SARS-CoV-2 nucleocapsid (N) protein. Antigen-specific cells were detected by monitoring the same functions as for the S protein. Strikingly, we observed no significant differences between the antigen exposure groups with and without breakthrough infections indicating that previously vaccinated individuals either do not develop N-specific T cells or at very low levels. This was true for both CD4+ (Fig. 4d) and CD8+ (Fig. 4e) T cells. Regarding the overall SARS-CoV-2-specific T cell response (not discriminating the CD4+ and CD8+ T cell or their specificity for S or N proteins) 3xVacc group had significantly increased frequencies of cytotoxic (1.7-fold; *p* = 0.01) and IFNγ-expressing (2.1-fold; *p* = 0.012) cells compared to the 2xVacc group (Fig. 4f). The largest proportion of SARS-CoV-2-specific cytotoxic T cells (discriminated were CD4+ S-specific, CD8+ S-specific, CD8+ N-specific, and CD4+ N-specific T cells) represented CD4+ S-specific T cells for 2xVacc, 2xVacc+α, 2xVacc+δ, and 3xVacc groups (*p* < 0.01 for all comparisons). The same cell population prevailed among the IFNγ-expressing cells (*p* < 0.05 for all comparisons) and IL-2-expressing cells (*p* < 0.0001 for all comparisons) for all antigen exposure groups. Similarly, in the case of TNFα-expressing T cells, CD4+ cells specific for the S-protein represented the large majority of the response for all (*p* < 0.05 for all comparisons) except the 2xVacc+δ group. Comparing the antigen exposure groups

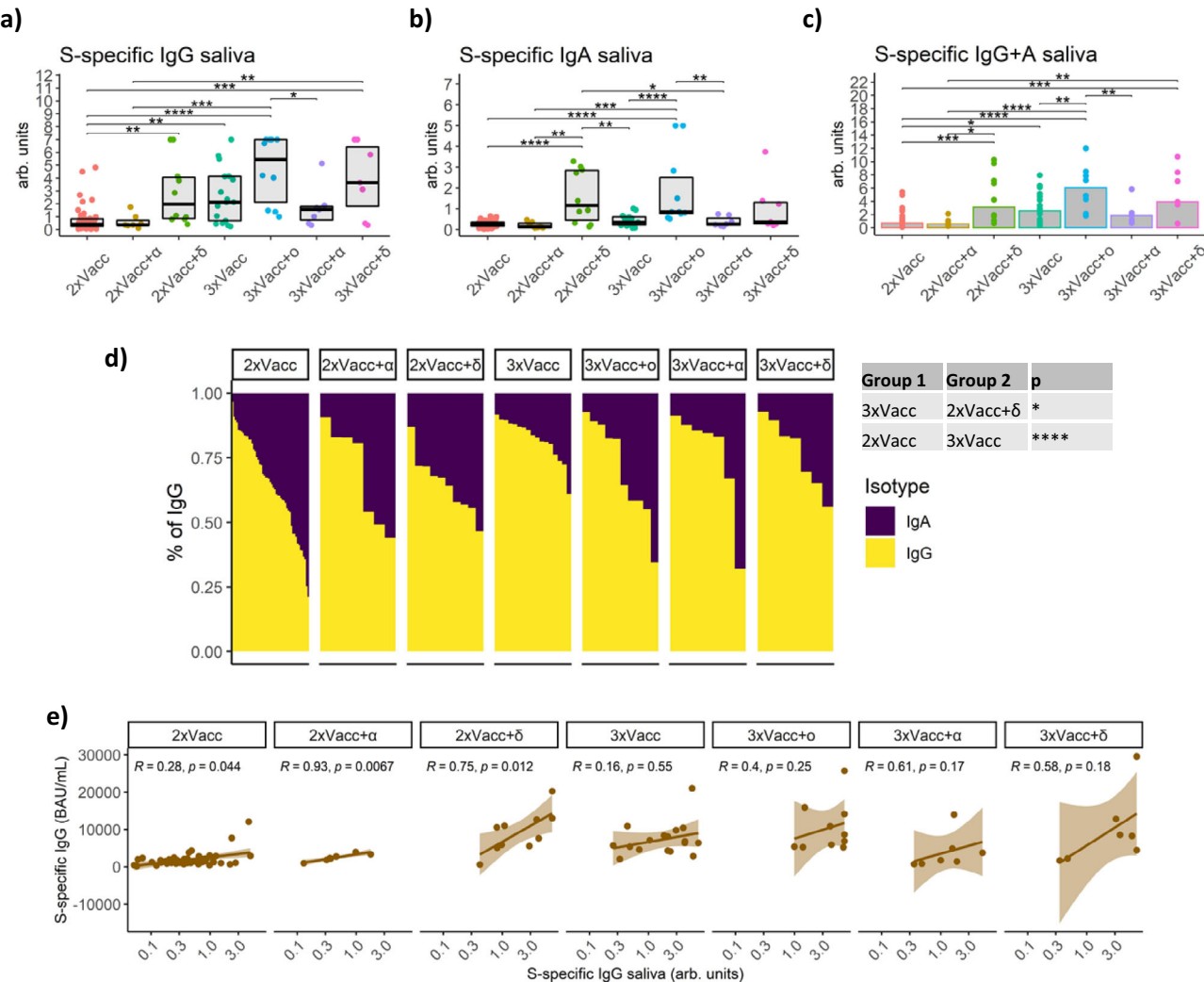

**Fig. 2 | SARS-CoV-2-specific antibodies in saliva.** S1-specific **a** IgG (exact $p$ values in sequential order from the upmost bracket: **$p$ = 0.0091, ***$p$ = 0.0003, *$p$ = 0.015, ***$p$ < 0.00012, ****$p$ < 0.0001, **$p$ = 0.0032, **$p$ = 0.0091), **b** IgA (exact $p$ values in sequential order from the upmost bracket: **$p$ = 0.0015, *$p$ = 0.019, ****$p$ < 0.0001, ***$p$ < 0.00027, ****$p$ < 0.0001, **$p$ = 0.0027, **$p$ = 0.0044, ****$p$ < 0.0001), and **c** IgG +IgA (exact $p$ values in sequential order from the upmost bracket: **$p$ = 0.0098, ***$p$ = 0.00041, **$p$ < 0.0015, **$p$ < 0.0023, ****$p$ < 0.0001, ****$p$ < 0.0001, *$p$ < 0.017, *$p$ < 0.016, ***$p$ < 0.00034) levels in saliva for the seven groups with different antigen exposure histories. Data is displayed as boxplots (**a**, **b**), indicating the first quartile, median, and third quartile, or bars showing the median (**c**) with individual data points. **d** Relative proportions of IgA and IgG isotypes among the S1-specific antibodies in the saliva of individuals with different antigen exposure histories.

Data are shown as a stacked bar plot for each individual. Statistically significant differences between the groups are indicated in the table next to the graph (*$p$ = 0.011, ****$p$ < 0.0001). **e** Correlations between the plasma and salivary S1-specific IgG for the seven antigen exposure groups. The 95% confidence intervals around the line of best fit are displayed as shading. The $r$ and $p$ values are given for each line. The following numbers of biologically independent samples were included in each group for all the graphs in this figure: 2xVacc, $n$ = 51; 2xVacc+α, $n$ = 7; 2xVacc+δ, $n$ = 10; 3xVacc, $n$ = 17; 3xVacc+o, $n$ = 10; 3xVacc+α, $n$ = 7; 3xVacc+δ, $n$ = 7. Differences between the groups were assessed using the two-sided Mann–Whitney test with Holm's correction for multiple testing. The strength of correlations was assessed by the two-sided Spearman's correlation test. Source data are provided as a Source data file.

no significant differences in proportions of the four cell populations were observed for any of the functions (Fig. 4g). Most of the study participants had detectable S-specific CD4+ T cells for all of the measured functions (Fig. 4h). Particularly high proportions were observed within the 3xVacc, 3xVacc+o, and 3xVacc+δ groups. The second most frequent response was CD8+ T cells specific for the S-protein. Here, the 3xVacc+o, 3xVacc+α, and 3xVacc+δ groups harbored the highest percentage of responders. More rare were individuals with detectable N-specific CD4+ and CD8+ T cells, especially within the 2xVacc and 2xVacc+α groups (Fig. 4h).

To sum up, additional antigen exposures of twice-vaccinated individuals do not significantly boost the frequencies of SARS-CoV-2-specific T cells except in the case of three vaccine doses. CD4+ T cell responses were more frequent than CD8+ T cells and more T cells were specific for S than N protein. Individuals with

omicron breakthroughs had an increased proportion of N-specific T cells.

**Individuals with breakthrough infections have a better correlated adaptive immune response**

The quality of antigen exposure is not only dependent on the magnitudes of individual immune components but also on their coordination. Apart from antibodies, many studies have demonstrated the importance of SARS-CoV-2-specific memory B and T cells[28–30]. Considering their distinct mechanism of action a multilayer immune response might be more effective at preventing SARS-CoV-2 infection.

We, therefore, correlated the measured immune parameters including antibody, memory B cell, and T cell responses for each of the seven groups with different antigen exposure histories. For the 2xVacc and 3xVacc groups, we observed a moderate to low degree of

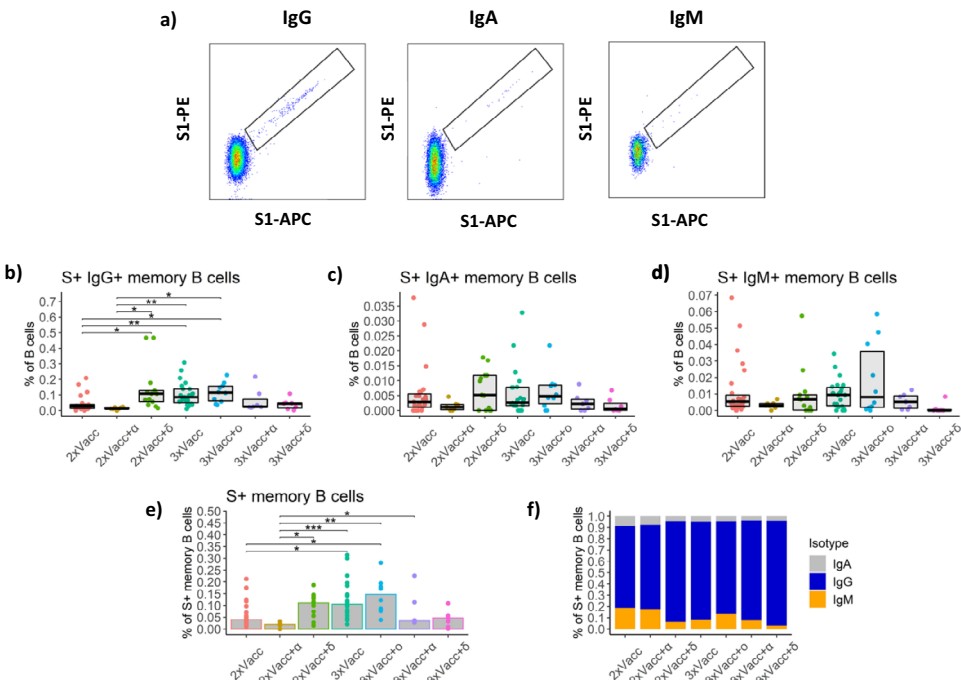

**Fig. 3 | SARS-CoV-2-specific memory B cell response after different combinations of vaccination and breakthrough infection. a** Representative flow cytometry pseudocolor plots for detection of S1-specific memory B cells with different BCRs. For the detailed gating strategy see Supplemental Fig. 2. Frequencies of **b** IgG + (exact *p* values in sequential order from the upmost bracket: *p = 0.014, **p = 0.005, *p = 0.015, *p = 0.014, **p < 0.004, *p = 0.047), **c** IgA+, **d** IgM+, and **e** total S1-specific memory B cells (exact p values in sequential order from the upmost bracket: *p = 0.014, **p = 0.005, *p = 0.015, *p = 0.014, **p < 0.004, *p = 0.047) in peripheral blood of individuals belonging to the seven antigen exposure groups expressed as a percentage of B cells. The data is displayed as boxplots (**b–d**), indicating the first quartile, median, and third quartile, or bars showing the median (**e**) with individual data points. **f** Relative proportions of S1-specific memory B cells bearing BCRs of a different isotype (color-coded) presented as stacked bar plots. Each section of the bar represents the median proportion of an isotype. The following numbers of biologically independent samples were included in each group for all the graphs in this figure: 2xVacc, n = 27; 2xVacc +α, n = 7; 2xVacc+δ, n = 13; 3xVacc, n = 23; 3xVacc+o, n = 10; 3xVacc+α, n = 7; 3xVacc +δ, n = 7. Differences between the groups were assessed using the two-sided Mann–Whitney test with Holm's correction for multiple testing. Source data are provided as a Source data file.

correlation among the parameters defining antibody response and among the parameters defining the T cell response. The rest of the immune response was poorly correlated (Fig. 5). For the rest of the antigen exposure groups, all with breakthrough infections, we observed a strong to moderate degree of correlation within the antibody compartment, and also the T cell compartment. For the 3xVacc +α and 3xVacc+δ groups, a considerable proportion of correlations among the T cell parameters was inverse (Fig. 5). Strong correlations between the antibodies and T cells and antibodies and memory B cells were mostly observed in the cases of the 2xVacc+δ, 3xVacc+δ, 2xVacc +α, and 3xVacc+α groups. Moreover, groups with breakthrough infections showed a higher degree of correlation between the memory B cell and T cell parameters compared to only vaccinated groups. Of note, 3xVacc+o, 3xVacc+δ, and 3xVacc+α groups had a high proportion of inverse correlations (Fig. 5). The parameters defining memory B cell response were weakly correlated regardless of antigen exposure group (Fig. 5).

Taken together, our data suggest a better correlated immune response among individuals with breakthrough infection when compared to those only vaccinated. Moreover, individuals with four antigen exposures had a higher proportion of inverse correlations.

## Discussion

There is increasing evidence that multiple antigen exposures might be needed for robust immunity against SARS-CoV-2[16,18,20,23]. Furthermore, studies have suggested that hybrid immunity in terms of vaccination and infection offers improved protection from the disease[33,34]. It is, however, not clear how different combinations of vaccination and breakthrough infection with SARS-CoV-2 variants shape the immune response. Here we systematically compared the antibody, memory B cell, and T cell responses of individuals that initially received 2 doses of mRNA vaccine and were later boosted by either breakthrough infection, vaccination, or both. We discriminated the infections with alpha, delta, and omicron variants. Our findings suggest augmented immune responses among twice-vaccinated individuals that recovered from the delta variant infection and thrice-vaccinated individuals that recovered from omicron breakthrough infection.

The assessment of SARS-CoV-2 immunity most often relies on the measurement of spike-specific antibodies in plasma[24]. We have shown that, following the initial two doses of mRNA vaccine, a breakthrough infection with delta, or a third vaccine dose with or without additional omicron or delta breakthrough significantly boosts the production of spike-specific and neutralizing antibodies. Conversely, the alpha breakthrough infection did not significantly enhance antibody production. Furthermore, we have shown that the group with delta variant breakthrough infection most efficiently neutralized delta SARS-CoV-2, and the group with omicron breakthrough most efficiently neutralized the omicron virus. This suggests that the pre-existing immunity is shaped by the variant causing breakthrough infection. Further supporting this hypothesis, individuals who recovered from the infection with a particular variant showed a stronger correlation between the spike-specific antibody levels and neutralization against that variant. In line with previous studies, the currently circulating omicron variant was significantly more resistant to neutralization than wild-type or delta and delta was more resistant to neutralization than wild-type for vaccinated individuals only[35,36]. The only exception was the group with the omicron breakthrough that equally neutralized all three variants. Overall, these findings suggest superior humoral immunity against

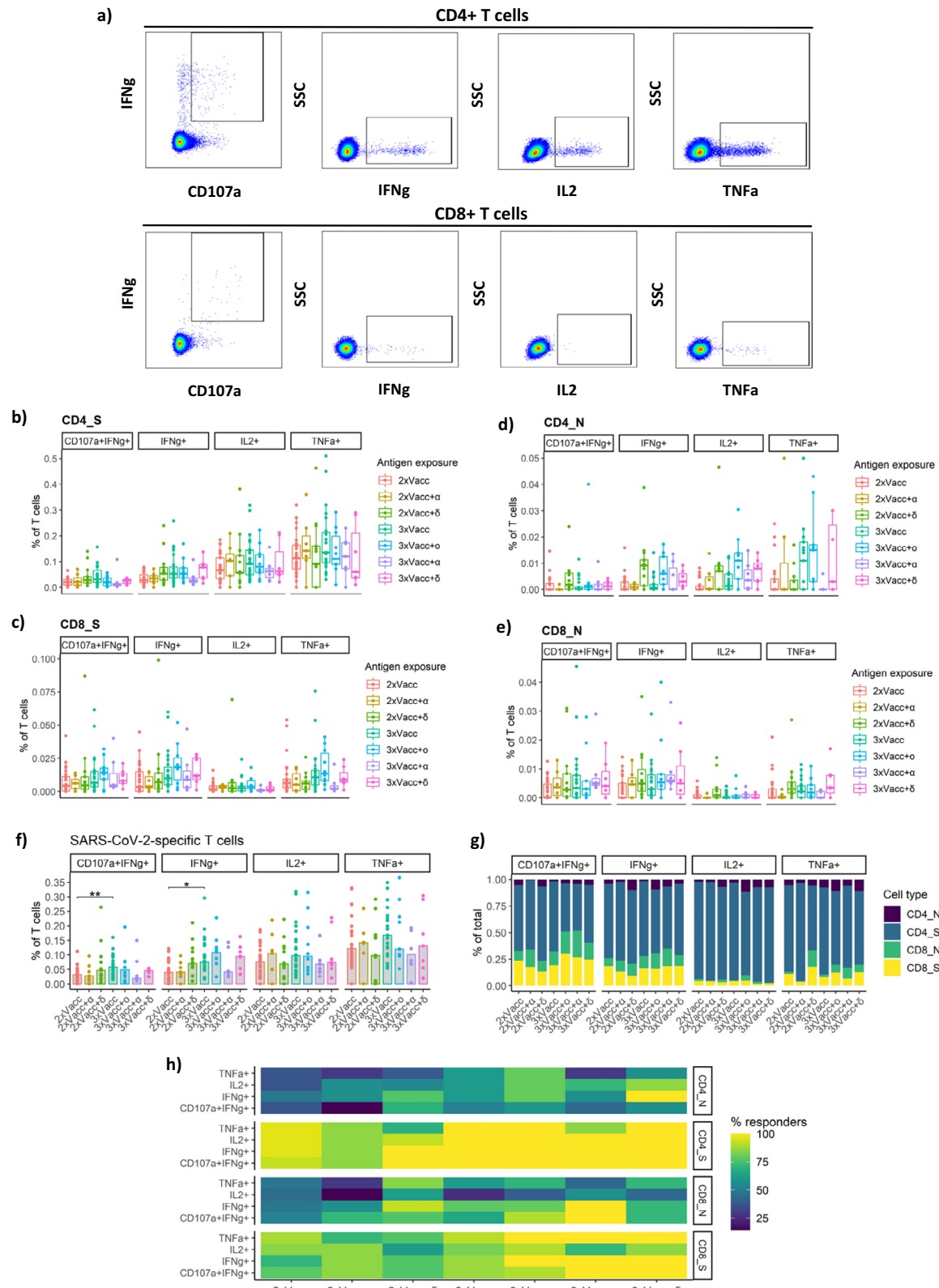

SARS-CoV-2 for vaccinated individuals with delta or omicron breakthrough infections.

When evaluating protection from SARS-CoV-2 infection, the measurement of antibodies in saliva might be more informative since they can neutralize the virus immediately after it enters the upper respiratory tract[26,27]. We demonstrated increased levels of salivary anti-spike antibodies for the same antigen exposure groups as in the case of

plasma IgG. Exceptionally high levels of spike-specific IgG and IgA in saliva were observed for the three times vaccinated individuals that recovered from omicron infection. This may be due to the increased replication of omicron in the upper respiratory tract compared to the previous variants and consequently stronger stimulation of the local mucosal immunity[37,38]. Moreover, given the abundance of IgG in comparison to IgA in the saliva our data demonstrate that the majority

**Fig. 4 | SARS-CoV-2-specific T cell response to different antigen exposure scenarios.** Frequencies of S-specific **a** representative flow cytometry pseudocolor plots demonstrating the detection of SARS-CoV-2-specific CD4+ and CD8+ T cells with different effector functions; cytotoxicity (CD107a and IFNγ expression), IFNγ expression, IL-2 expression, and TNFα expression. For the detailed gating strategy see Supplemental Fig. 3. **b** CD4+ T cells and **c** CD8+ T cells with different effector functions as a percentage of bulk T cells. Frequencies of N-specific **d** CD4+ T cells and **e** CD8+ T cells with different effector functions as a percentage of bulk T cells. Different antigen exposure groups are color-coded. For panels **b**−**e**, the data is displayed as box-whisker-plots, indicating minimum, first quartile, median, third quartile, and maximum, with individual data points. **f** Frequencies of S or N-specific T cells with different functions, not discriminating the CD4+ and CD8+ T cells. The exact *p* values in sequential order from the leftmost bracket: **p = 0.01, *p = 0.012.

Data is displayed as bars showing the median with individual data points. **g** Relative proportions of S-specific CD4+, N-specific CD4+, S-specific CD8+, and N-specific CD8+ T cells performing different functions for each antigen exposure group. Data are presented as stacked bar plots. CD4+/CD8+ T cells and their specificities for N or S proteins are color-coded. **h** Percentage of individuals with detectable SARS-CoV-2-specific CD4+ and CD8+ T cells within each group. The percentage of responders (individuals where the frequency of stimulation-responding T cells was higher than in the negative control) is color-coded. The following numbers of biologically independent samples were included in each group for all the graphs in this figure: 2xVacc, *n* = 27; 2xVacc+α, *n* = 7; 2xVacc+δ, *n* = 13; 3xVacc, *n* = 23; 3xVacc+o, *n* = 10; 3xVacc+α, *n* = 7; 3xVacc+δ, *n* = 7. Differences between the groups were assessed using the two-sided Mann–Whitney test with Holm's correction for multiple testing. Source data are provided as a Source data file.

of the spike binding activity in saliva was by IgG and not IgA antibodies regardless of the antigen exposure history. The highest proportion of IgA spike binding was among the twice-vaccinated individuals, while the third vaccination did not lead to a further increase in IgA immunity. Interestingly, we observed that salivary spike-specific IgG does not correlate well with the plasma IgG levels for most of the antigen exposure groups, suggesting that a significant part of the salivary IgG is locally produced.

Following the SARS-CoV-2 infection or vaccination, the antibody levels rapidly decline increasing the chance of breakthrough infections[10,11,31]. This is, however, not true for the memory B cells that are more persistent and therefore particularly important for long-term protection against severe SARS-CoV-2 infection[28,31]. We found that most of the spike-specific memory B cells found in the peripheral blood of vaccinated individuals with or without breakthrough infections bear IgG BCR. The highest frequencies of those cells were observed for the groups with strong antibody responses except for the thrice vaccinated group that recovered from delta infection. A similar trend was observed for the IgA-expressing spike-specific memory B cells but was not statistically significant due to the large intraindividual variability. Together with the antibody measurements, the memory B cell data implies that the fourth antigen exposure does not further augment the B cell immunity established after three antigen exposures, except in the case of the breakthrough with a genetically distinct omicron variant.

Besides antibodies, T cells represent an important mechanism for limiting viral infections and were previously associated with protection from SARS-CoV-2 infection[29,30]. Importantly, the frequency of these cells remains at elevated levels for a longer time after infection or vaccination than the antibody level[31,39]. Our data show that further antigen exposures, either by infection or vaccination, of the twice-vaccinated individuals, mostly do not improve the SARS-CoV-2-specific T-cell response. The only exception was the thrice vaccinated group that had significantly elevated cytotoxic and IFNγ-secreting SARS-CoV-2-specific T cells. Of note, SARS-CoV-2-specific CD4 + T cells were considerably more frequent than the CD8 + T cells and T cells specific for the spike protein prevailed over those recognizing the nucleocapsid protein. This is in contrast to the only infected individuals that develop equal levels of nucleocapsid- and spike-specific CD4 + T cells as observed in our previous studies[31,32]. Moreover, the frequency of these cells is considerably higher in only infected individuals suggesting that vaccination impairs the formation of nucleocapsid-specific T cells possibly due to the reduced viral replication and availability of the antigen. Interestingly, some of the vaccinated individuals without breakthrough infection had detectable nucleocapsid-specific T cells which could be explained by the high levels of pre-existing naïve or cross-reactive T cells previously documented among SARS-CoV-2 naïve individuals[30,40]. To sum up, individuals boosted by infection or vaccination do not have significantly augmented T cell immunity compared to the subjects that received only two vaccine doses.

The quality of immunity against viral infections does not only depend on a single but rather on the synchronization of multiple immune mechanisms. We demonstrated that individuals with breakthrough infections have better correlated parameters defining the adaptive immune response than those only vaccinated. Increased coordination of the immune response after breakthrough infection complies with the findings demonstrating better protection among individuals with hybrid immunity[33,34].

Taken together we compared the adaptive immune response of individuals with different antigen exposure histories in terms of vaccination and breakthrough infection. Our findings suggest that delta but not an alpha breakthrough infection or third vaccination of doubly vaccinated individuals considerably improves SARS-CoV-2 immunity. Strikingly, the fourth antigen exposure did not further augment the immune response compared to the three antigen exposures except for the omicron breakthrough infection. The observed differences between the groups with different antigen exposure scenarios are likely due to a combination of factors such as antigenic distance, the severity of breakthrough infection, the immunogenicity of the SARS-CoV-2 variants, and the time passed between the antigen exposures. No biases towards high-risk populations (elderly, immunosuppressed) were identified for any of the groups. Limitations of this study are that it is observational and does not reveal the mechanism driving improved immune response in particular groups. The study participants were employed at the University Hospital Bonn, therefore, the cohort was biased toward higher-educated individuals, aged between 19 and 69 years which likely does not reflect the general population. Moreover, we did not measure the neutralization capacity against the alpha variant, and some of the compared groups have relatively low sample numbers. Strengths of the study are that the participants were monitored for SARS-CoV-2 infections by RT-PCR and self-testing throughout the pandemic and that the SARS-CoV-2 variants were identified by sequencing rather than the time point of infection.

## Methods
### Study cohort
A total of 110 individuals that were initially immunized with 2 doses of mRNA-based SARS-CoV-2 vaccine and subsequently infected and/or vaccinated were recruited for the study. The recruitment was conducted by the occupational healthcare department of the University Hospital Bonn. The first contact was established by telephone after which a written invitation and a consent form were sent to each participant. All individuals were sampled 2−9 weeks following the last antigen exposure. The individuals belonging to different study arms were preselected so that the times from the last antigen exposure did not significantly differ between the groups. Age or sex was not among the selection criteria and no significant differences in age and sex distribution were observed between the groups. Detailed information on the antigen exposure and sampling time points as well as demographic information is provided in Supplemental Fig. 1 and Supplemental Table 1. Breakthrough infections were confirmed by RT-PCR

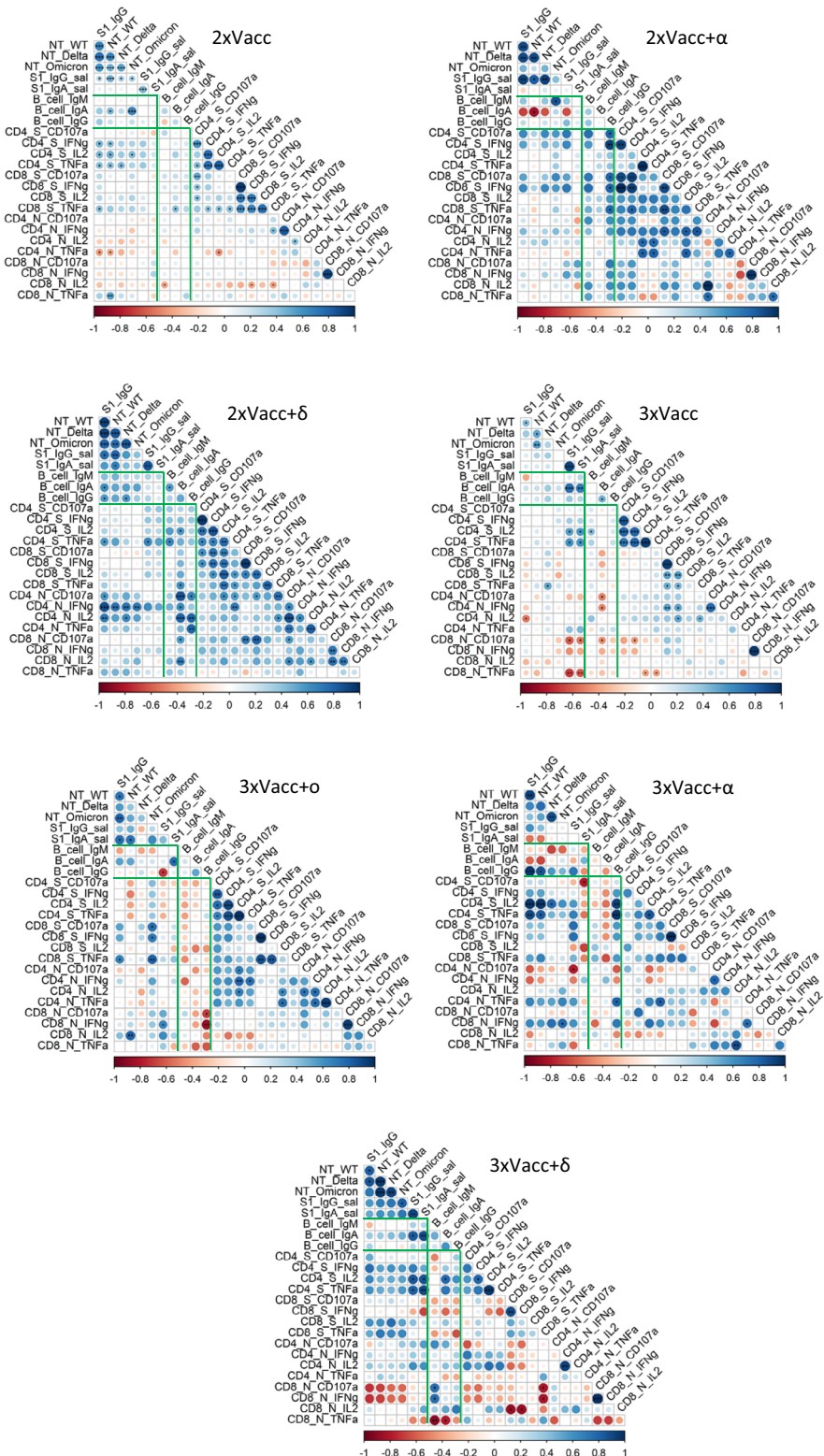

**Fig. 5 | Correlation between the parameters of the adaptive immune response to different antigen exposure scenarios.** Correlation matrices demonstrate the strength of correlations between the measured immune parameters for each of the antigen exposure groups. The strength of a correlation (Spearman's correlation coefficient) is depicted by the size and color of the circle, significance is indicated by asterisks. The exact *p* values are given in the Source data file. Green lines separate correlations between different branches of the measured immune parameters; antibodies, memory B cells, and T cells. The following numbers of biologically independent samples were included in each group for all correlated parameters: 2xVacc, *n* = 54 (for plasma antibodies), 2xVacc, *n* = 51 (for plasma antibodies), 2xVacc, *n* = 27 (for B and T cells); 2xVacc+α, *n* = 7; 2xVacc+δ, *n* = 13 (for plasma antibodies, B cells, and T cells), 2xVacc+δ, *n* = 10 (for saliva antibodies); 3xVacc, *n* = 23 (for plasma antibodies, B cells, and T cells), 3xVacc, *n* = 17 (for saliva antibodies); 3xVacc+o, *n* = 10; 3xVacc+α, *n* = 7; 3xVacc+δ, *n* = 7. The strength of correlations was assessed by the two-sided Spearman's correlation test. Source data are provided as a Source data file.

and the viral RNA was sequenced as a part of routine SARS-CoV-2 variant monitoring at the diagnostics department of the Institute of Virology, University Hospital Bonn. All participants were either employed or studied at the University Hospital Bonn at the time of sampling but were not necessarily healthcare workers. As employees of the University Hospital Bonn study participants were obliged to perform two antigen tests every week and RT-PCR whenever they developed symptoms similar to Covid-19. Furthermore, individuals with a history of previous SARS-CoV-2 infection were not taken into the study. All individuals that had a breakthrough infection were infected only once. For the groups without breakthrough infections, only individuals without confirmed SARS-CoV-2 infection, and negative nucleocapsid ELISA results were included.

### Ethics approval
The study was approved by the Ethics Committee of the Medical Faculty of the University of Bonn (ethics approval numbers 125/21) and all participants provided written informed consent. No compensation was provided for the participants.

### Sample collection and storage
Study participants provided peripheral blood specimens, saliva, and pharyngeal swabs. Blood was centrifuged and EDTA-plasma was stored until analysis ($-80\,°C$). Before saliva collection participants were instructed not to eat or drink for at least 60 min. Afterward, participants would retain saliva for 1–2 min and expectorate it in a centrifuge tube. Saliva samples were centrifuged to remove solid particles and frozen at $-20\,°C$. PBMC were isolated by density gradient centrifugation and cryopreserved in liquid nitrogen.

### Determination of SARS-CoV-2 S- and N-specific antibodies in plasma
N-specific antibody levels were assessed using the Roche Cobas® SARS-CoV-2 assay following the manufacturer's protocol. For the determination of S1-specific IgG, an in-house quantitative ELISA was used. For that microtiter plates with high-binding capacity were coated with $100\,\mu l$ of coating buffer (carbonate-bicarbonate buffer, pH = 9.6) containing $1\,\mu g/ml$ of recombinant SARS-CoV-2 S1 protein (Biotinylated SARS-CoV-2 (COVID-19) S1 protein, Acrobiolabs, S1N-C82E8-200ug-AC). Coated plates were covered and incubated overnight at 4 °C. After washing with wash buffer (PBS with 0.05% (v/v) Tween®−20) plates were blocked (PBS containing 1% (w/v) BSA) to prevent unspecific binding. Cryopreserved EDTA plasma samples were thawed and diluted 1:3200 in the blocking buffer. Blocked plates were washed, incubated with plasma and standard samples, washed again, and incubated with $100\,\mu l$ HRP-conjugated anti-IgG antibody (Goat anti-Human IgG (Heavy chain) Secondary Antibody, HRP, Invitrogen, A18805) diluted 1:8000 in wash buffer. Incubation steps were performed for 1 h at 37 °C. Afterward, plates were washed and $100\,\mu l$ of the substrate solution was added (TMB Chromogen Solution, Life technologies, 002023). The reaction took place at room temperature for 5 min until the addition of $50\,\mu l$ of $0.2\,M\,H_2SO_4$. Finally, optical density at 450 nm was measured. The background-subtracted $OD_{450}$ readings were interpolated to the standard dilution curve calibrated to the international WHO standard (NIBSC reference number: 20/136). The positivity cutoff was determined based on measurements of plasma samples from healthy individuals collected before the Covid-19 outbreak. All samples were measured in duplicates.

### Determination of SARS-CoV-2 S-specific IgG and IgA in saliva
The relative amounts of S1-specific IgA and IgG in saliva were measured by in-house quantitative ELISA. High-binding microtiter plates were coated with $100\,\mu l$ of coating buffer (carbonate-bicarbonate buffer, pH = 9.6) containing $1\,\mu g/ml$ of recombinant SARS-CoV-2 S1 protein (Biotinylated SARS-CoV-2 (COVID-19) S1 protein, Acrobiolabs, S1N-C82E8-200ug-AC). After overnight incubation at 4 °C the plates were washed (PBS with 0.05% (v/v) Tween®−20), blocked (PBS containing 3% (w/v) BSA), and washed again. Frozen saliva samples were thawed, diluted in sample buffer (PBS containing 1% (w/v) BSA), and pipetted onto the plate. Following incubation with saliva samples and standard dilutions plates were washed and incubated with $100\,\mu l$ HRP-conjugated anti-IgG antibody (Goat anti-Human IgG (Heavy chain) Secondary Antibody, HRP, Invitrogen, A18805) diluted 1:8000 in wash buffer or $100\,\mu l$ HRP-conjugated anti-IgA antibody (Goat anti-Human IgA (Heavy chain) Secondary Antibody, HRP, Invitrogen, A18781) diluted 1:1000 in wash buffer. Plates were then washed and $100\,\mu l$ of the substrate was added (TMB ELISA Substrate, High Sensitivity, Abcam, ab171523). The reaction took place at room temperature for 5 min, followed by the addition of $100\,\mu l$ of $1\,M\,H_2SO_4$. Finally, optical density at 450 nm was measured. The background-subtracted $OD_{450}$ readings were interpolated to the standard dilution curve. The cutoff for positivity was determined based on measurements of saliva samples from healthy individuals collected before the Covid-19 pandemic. The same concentrations (15 ng/ml) of S1-specific monoclonal antibody (anti-SARS-CoV-2-RBD antibody, clone CR3022, Abcam, ab278112/ab273073) with IgA or IgG constant region was measured to make the $OD_{450}$ readings comparable between the assays. All samples were measured in duplicates.

### Plaque reduction neutralization assay
The neutralization capacity of plasma samples was determined by a plaque reduction neutralization assay. Therefore, plasma was heat-inactivated for 30 min at 56 °C and serially two-fold diluted in OptiPRO SFM (Gibco, 12309-019) cell culture medium. A total of 10 dilutions between 2-fold and 32768-fold were measured for each sample depending on the neutralization capacity of a specimen. No further technical replicates were performed. Each dilution was combined with 80 plaque-forming units of SARS-CoV-2 (either wild-type, delta, or omicron variant) in OptiPRO SFM (Gibco, 12309-019) cell culture medium, incubated for 1 h at 37 °C, and added to Vero E6 cells (ATCC, CRL-1586). The cells were seeded in 24-well plates at $1.25 \times 10^5$ cells/well 24 h earlier. Following 1 h incubation at 37 °C, the inoculum was removed and cells were overlaid with a 1:1 mixture of 1.5% (w/v) carboxymethylcellulose in 2xMEM supplemented with 4% FBS. After incubation at 37 °C for three days, the overlay was removed and the 24-well plates were fixed using a 6% formaldehyde solution and stained with 1% crystal violet in 20% ethanol revealing the formation of plaques. The number of plaques was plotted against the serum/supernatant dilutions and IC50 was determined using GraphPad Prism software version 9.4.1 (681).

### B cell isolation
B cells were enriched from cryopreserved PBMC samples by immunomagnetic isolation (REAlease® CD19 MicroBead Kit, human, Miltenyi Biotec, 130-117-034). Isolation was performed following the manufacturer's instructions. Briefly, PBMCs, which had been thawed and rested overnight, were resuspended in recommended isolation buffer and labeled with anti-CD19 antibodies coupled to magnetic beads. Labeled cells were then immobilized onto a magnetic column. B cell-depleted flow-through was used for the assessment of CD4 + T cell responses. Immobilized B cells were washed out of the column and enzymatically released from magnetic beads.

### Detection of S1-specific memory B cells by flow cytometry
SARS-CoV-2 S1-specific B cells were identified by immunofluorescent tagging with recombinant wild-type SARS-CoV-2 S1 protein. Therefore, the magnetically isolated B cells were resuspended in FACS buffer (PBS supplemented with 2% FCS, 0.05% NaN₃, and 2 mM EDTA) and incubated with the fluorescently labeled recombinant SARS-CoV-2 S1 protein (Biotinylated SARS-CoV-2 (COVID-19) S1 protein, Acrobiolabs, S1N-

C82E8-200ug-AC). To minimize the unspecific binding of the probe, S1 protein was conjugated to two different streptavidin-fluorochrome conjugates, streptavidin-PE (Biolegend, 405204) and streptavidin-APC (Biolegend, 405207), in an equimolar ratio. After 15 min at 4 °C anti-IgG-BV421 antibody (clone G18-145, Biolegend, 562581, diluted 1:20) was added to the cell suspension and the incubation was continued for another 15 min. Following the binding of S1 probes, cells were washed with PBS and stained for viability (ZombieAqua, Biolegend, 423102) for 15 min at 4 °C. Subsequently, cells were washed with FACS buffer and incubated with a solution of antibodies blocking human Fc receptors (FcR block, Miltenyi Biotec, 130-059-901, diluted 1:10). After 10 min a mixture of fluorescently labeled antibodies binding to surface antigens of B cells was added. The mixture included the following fluorescently-labeled antibodies: anti-CD3-BV510 (clone UCHT1, Biolegend, 300448, diluted 1:40), anti-CD27-BV605 (clone O323, Biolegend, 302830, diluted 1:20), anti-IgM-BV785 (clone MHM-88, Biolegend, 314544, diluted 1:20), anti-IgA-VioBright 515 (clone REA1014, Miltenyi Biotec, 130-116-886, diluted 1:40), anti-CD21-PE-Cy7 (clone Bu32, Biolegend, 354912, diluted 1:160), and anti-CD19-APC-Cy7 (clone HIB19, Biolegend, 302218, diluted 1:80). The staining was performed at 4 °C for 15 min. Following incubation, the cells were washed again and acquired on a BD FACS Celesta flow cytometer with BD FACSDiva™ Software Version 8.0 (BD Bioscience). Possible longitudinal fluctuations in laser intensity were monitored every day before the experiment and were compensated using fluorescent beads (Rainbow beads, Biolegend, 422905). The data were analyzed with the FlowJo Software version 10.0.7 (TreeStar). The frequency of S-specific memory B cells was calculated by subtracting the average frequency of S-binding memory B cells in eight healthy donor samples collected before the outbreak of the SARS-CoV-2 pandemic. No technical replicates were performed due to the scarcity of the samples.

### Ex vivo stimulation of T cells
Overnight-rested B-cell-depleted PBMC were seeded in 96-well U bottom plates and stimulated with wild-type SARS-CoV-2 PepTivator (Miltenyi Biotec, 130-127-951/130-126-698) overlapping peptide pools spanning the entire sequences of SARS-CoV-2 S or N proteins, in presence of anti-CD107a-APC (clone H4A3; Biolegend, 328620, diluted 1:40) antibody. One million cells were stimulated per condition and the final concentration of each peptide was 1 µg/ml for both peptide pools. Co-stimulatory antibodies (BD FastImmune™ CD28/CD49d, BD Bioscience, 347690) were added to a final concentration of 1 µg/ml. Stimulation was performed at 37 °C for 6 h. For each sample, an equally treated DMSO-stimulated negative control was included. As a positive control, cells were stimulated with PMA (20 ng/ml) (Sigma-Aldrich, P1585-1MG) and ionomycin (1 µg/ml) (Sigma-Aldrich, I3909-1ML). One hour into stimulation Golgi Stop (BD Bioscience, 554724) and Golgi Plug (BD Bioscience, 555029) were added (final concentration 1 µg/ml) to inhibit vesicular transport and prevent the secretion of the cytokines.

### Detection of SARS-CoV-2-specific T cells by flow cytometry
Following stimulation, cells were washed with PBS and stained with Zombie Aqua (Biolegend) dye to discriminate viable cells. The staining was performed for 15 min at 4 °C. Subsequently, samples were washed with FACS buffer, fixed, and permeabilized in CytoFix/CytoPerm Solution (BD Bioscience, 554714) for 15 min at 4 °C. Fixed cells were then washed with 1x Perm/Wash Buffer (BD Bioscience, 554723), and stained for intracellular markers for 15 min at 4 °C using the following antibody conjugates; anti-CD3-APC-Cy7 (clone UCHT1, Biolegend, 300426, diluted 1:40), anti-CD4-BV786 (clone SK3, BD Bioscience, 344642, diluted 1:40), anti-IFNγ-PE (clone B27, Biolegend, 506507, diluted 1:40), anti-TNFα-BV421 (clone Mab11, Biolegend, 502932, diluted 1:80), and anti-IL2-AF488 (clone MQ1-17H12, Biolegend, 500304, diluted 1:20). Each antibody was checked for performance and titrated before use. Finally, cells were washed with PBS and acquired on a BD FACS Celesta with BD FACSDiva™ Software Version

8.0 (BD Bioscience). Frequencies of antigen-specific CD4 + T cells were calculated as negative-control-subtracted data. Possible longitudinal fluctuations in laser intensity were monitored daily before the experiment using fluorescent beads (Rainbow beads, Biolegend, 422905). If needed PMT voltages were adjusted to ensure constant signal intensity over time. The data were analyzed with the FlowJo Software version 10.0.7 (TreeStar). No technical replicates were performed due to the scarcity of the samples.

### Statistical analysis
Statistical analysis was performed using RStudio 2021.09.0 Build 351 software[41]. Differences between the groups were assessed using the Mann–Whitney test or Wilcoxon test for matched data with Holm's correction for multiple testing. All tests were performed two-sided. The strength of correlations was evaluated by Spearman's test. Statistical significance is indicated by the following annotations: *$p < 0.05$, **$p < 0.01$, ***$p < 0.001$, ****$p < 0.0001$.

### Reporting summary
Further information on research design is available in the Nature Portfolio Reporting Summary linked to this article.

## Data availability
The data contain information that could compromise the privacy of research participants. Data sharing restrictions imposed by national and transnational data protection laws prohibit the general sharing of data. However, upon submission of a proposal to the corresponding author and approval of this proposal by (i) the principal investigator, (ii) the Ethics Committee of the University of Bonn, and (iii) the data protection officer of the University Hospital Bonn, data collected for the study can be made available to other researchers. A source data file containing the statistics presented in the figures and a Supplemental table containing demographic information are provided with this paper. Source data are provided with this paper.

## Code availability
No custom code or mathematical algorithm was generated for this study.

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

## Acknowledgements

The study was supported financially by the State of North Rhine-Westphalia, and the Viral NRW Network (grant number: CPS-1-1C acquired by H.S.). No other financial support by any third parties was received. We thank the participants of this study that generously provided their samples.

## Author contributions

Conceptualization: J.P., W.M.P., and H.S.; methodology: J.P., G.A., and H.S.; investigation: J.P., W.M.P., K.P., J.Z., M.B., C.B.S., and H.P.; resources: J.P. and W.M.P.; writing—original draft: J.P.; writing—review & editing: J.P., H.S., and G.A.; funding acquisition: J.P. and H.S.; supervision: H.S.

## Funding

## Competing interests

The authors declare no competing interests. The idea, the plan, the concept, the protocol, the conduct, the data analysis, and the writing of the manuscript of this study were independent of any third parties, including the funding agency.
