## [Peer Review File · Nature Communications]

REVIEWER COMMENTS

Reviewer #1 (Remarks to the Author):

In the manuscript "SARS-CoV-2 immunity following different combinations and breakthrough infections" Pusnik et al. investigate the adaptive immune response after different combinations of vaccinations (two vs. three doses) in the presence or absence of breakthrough infection. In addition, they also investigate the immune response after breakthrough infection with different variants of concerns (alpha, delta, and omicron). A scenario very much reflected in current real life. Importantly, the variant of concern was determined by RT-PCR rather than inferring it from the time point of infection. Overall, this is a very important and timely study as this answers the question if the times of antigen exposure and/or the type of exposure (vaccination vs. natural infection) influences adaptive immune responses.

Minor comments:

- All samples were collected 2-9 weeks following the last immunization. Please clarify that this time point also refers to breakthrough infections. i.e., 2 weeks after infection.... It would be important to show that the average time point within these 2-9 weeks was comparable between groups, as the adaptive immune response can change significantly within this time point.
- It would be interesting to compare the 3+Vacc+O group to a group that has been vaccinated four times, as this would most accurately reflect the 4-time antigen exposure in both groups.
- Is it remarkable that previously vaccinated individual's to not develop T cell response against the N protein. It would be good to include results from individuals that have only been naturally infected and not vaccinated to demonstrate that N specific responses are detected in the absence of vaccination.

Reviewer #2 (Remarks to the Author):

The study by Pusnik et al presents a comprehensive immunogenicity combined vaccine with breakthrough infection study. However, there are concerns below about cohort detail that undermine the strength of the findings, as my main issue is that the timing of infection plays a bigger role than the antigenic difference but is not sufficiently described- is the timepoint of sampling after infection is comparable between groups (2-9 weeks?). The methods are sufficiently described. Some statistical correlations are not adequately described. FACS gating strategies and gate frequencies are not given.

Main comment:

1. Cohort details are needed for subject details as booster doses are recommended for at risk groups which may differ from vaccine only groups.
2. The timing between vaccination and infection is also likely different between the subjects infected with different variants and therefore their boosting potential is either related to timing post vaccination (i.e., infection at a short-term memory versus long term memory when neutralising antibodies have waned is likely to recruit different immune parameters) and/or antigenic distance and immune evasion from the ancestral vaccine. Furthermore, the sample timepoint after infection is not given (e.g. N positivity rates after alpha infection- is this due to long sample timepoint?) and 2-9 weeks after infection samples were compared?
3. Often immunisation is used to refer to vaccine or recovery from infection as a form of immunization. This is confusing for the reader as immunisation typically means vaccine. If you want to use the term this way throughout the article (including the way you refer to your different groups) then this needs to be flagged in the introduction for clarity to the reader. The difference between hybrid immunity (infection then vaccination) or breakthrough infection (vaccination then infection) and immunization does not equate to vaccination or recovery from infection but vaccination however is used interchangeably in

the text.

4. No information on surveillance for infection method for this study. The commentary by the authors in the results seem to suggest that all volunteers only had one infection i.e. alpha, delta or omicron (i.e. lines 225-226) but what infection surveillance was carried out? Line 64: State here what the study type was i.e. observational. There needs to be more information on how were infections detected. And how often volunteers were surveyed. Was it just if volunteers developed symptoms? Based on national surveillance? Through GP attendance? Individual/self-diagnosed symptom detection?

Minor comments:

Abstract: line 27, were T cell responses impacted by infection?

Introduction:

Line 40. introduction: Cut to just "However similar to infection-induced immunity..." and delete "relatively" (line 41)- as relatively to what?

Line 42. change to "By the time the first vaccines were rolled out/licenced/in use.." And add reference for this sentence.

Line 43. delete 'several'.

Line 46. add official Pango lineage variant classification not just "delta" i.e. B.1.617.2 (also for Alpha and Omicron infections).

Line 47. Less frequent than what?

Line 48. Current vaccines do not significantly limit transmission, but boosters are recommended to increase protection against severe disease.

Line 49, third and now fourth doses are offered in many developed countries. Update for wider relevance.

Line 51. add reference to support. Vaccines may have reduced vaccine efficacy against Omicron compared to the ancestral virus, but still maintain high level of protection against mortality.

Line 52- in which scenario does vaccine cause more severe disease? This is an anti-vaccine sentiment that is not supported by a reference or data. Update or remove.

Line 55/56 What is meant by 'equal immune response'? Antibody titre? T cell? More specific parameters are needed.

Line 57: better protected from disease severity or infection risk? Be specific

Line 59 Change to " However, there is heterogeneity in breakthrough infections amongst vaccinated individuals in terms of the number of previous immunizations and the variant that causes the infection". Needs a reference.

Line 61 got -> became .

Line 64 Change to "amongst individuals" Change "the" to "their" or delete.

Line 65 add in alpha variant nomenclature

Line 65-69 I would suggest lower case "vacc" and capitalise Omicron etc throughout.

Line 69: "In most cases?" What does this mean. Be specific i.e. was this significant or proportion of subjects? Do you mean most immune end points or most volunteers or most of the groups?

Line 71: This section should be moved to the discussion. What is meant by 'more robust immunity'? The magnitude of the response and at what timepoint following or during infection to impact outcomes of infection for severity? The timing between vaccination and infection is also likely different between the subjects infected with different variants and therefore their boosting potential is either related to timing post vaccination i.e. infection at a short term memory versus long term memory when neutralising antibodies have waned is likely to recruit different immune parameters) or antigenic distance and immune evasion from the ancestral vaccine. From Fig 1A, though minimal details on timepoint intervals are given.

Methods:

Line 75: "Individuals"? – cohort details are needed for age/gender/timepoints/controls and date of recruitment/study period.

What were the third and fourth vaccinations? Were these also mRNA vaccines? What determined if they had third and fourth vaccines or didn't? age groups or underlying conditions have been prioritised for boosters- the cohort should be specified.

Line 81: "were not necessarily healthcare workers." Is not sufficient cohort details.

Line 133: delete "the" add in version number of software (also needed in Reporting summary).

Line 150: pandemics -> pandemic.

Line 158: change to "as a positive control, cells.."

Line 170: change: "acquired on a FACS Celesta"

Results:

Demographic information needed included about the volunteers. Was there any underlying cohort differences between the 7 groups? The timepoint intervals for sampling should be specified on Figure 1A, range, mean + stdev. Was this different between the variants? See main point above about interval between infection and vaccination recruiting different immune responses.

What does Figure 1a represent? Did all volunteers get omicron infection at month 12?/ 3 months after their 3rd vaccination?

Fig1c and 1g for Omicron figures should have the same axis as the other two (scale 10-10,000 vs 100-10,000)

Fig1d: Legend needs to state what the overlaid numbers refer to – IC50 medians?

Line 183: revise title, "Not only the number but also the type of immunization is important for potent humoral immunity against the SARS-CoV-2"

The study only describes mRNA immunization, in this case again the authors term immunization with recovery from infection.

Line 185: And Ref, suggest Khoury Nat Med 2020.

Line 224: The authors say the N antibody data complies with/reflects the S1 specific data but where does it say what % volunteers seroconverted to S1-specific IgG? Fig1b shows significant differences but not number/% volunteers. This information could be inferred by counting the dots but it would be good to have this information specifically shown and/or the N antibody data presented in the same way as Fig1b.

Also the authors state that the baseline is the 2xvacc but it would be good to see the data presented against the pre-pandemic levels (WHO standard).

Line 253. Delete 'partially true', suggest: "In contrast to plasma, the only significant correlations in saliva were ..."

Fig2d: Y axis % of IgG -> should be Ig? How can IgG be less than 100% of itself? Are the colours flipped or is this proportion of Ig? Again need legends for these figures: e.g. Fig2e what does the shading indicate?

Line 263: The FACS gating strategy and gate frequency should be added as supplementary, especially for the definition of memory B cells (Fig 3a, 4a).

Line 269/270: I don't agree with this conclusion is correct for IgM ie 2xvaccdelta looks very similar to 2xvacc even with variability. I think best to state no statistical differences seen, though there was a trend with some vaccine combinations such as IgA 2xvaccdelta, 3vacc and put in discussion that this might be due to large intraindividual variability.

Line 277/278: Statistics needed for support.

Line 287: delete "the (change to "protection from severe disease").

Figure 4: Further detail needed in legends- median and IQR shown?

Line 297: need to be careful with wording given no significance. Say a trend not that they were higher.

Line 300: Again can't say it was highest as not statistically significant. Also need to tell us what the medians were.

Line 307-308: Is it meant that infection groups don't develop N-specific T cell responses?

Line 312-318: Stats needed to support.

Line 319: Put in ref to fig4h after "functions".

Line 326-327: Again these conclusions need stats to back them up (from above paragraph).

Figure 5 annotation: Did you use both Pearson's and Spearman's as the figure legend mentions both and the heatmap label is not titled with either.

Discussion:

Line 359: Given the timing of infection has not been disclosed, it is problematic to refer to B cell memory and T cell memory responses based on a sampling timepoint as memory rather than phenotype against Omicron/Delta etc. Unless the assertion is that these are cross-reactive vaccine induced memory B cells based on the timepoint of sampling (which is not specified). Or if the cells have been phenotyped for memory – please include gating strategy. Please clarify.

Line 362: "responses"

Line 364: change to "of SARS-CoV-2"

Ref for this statement (line 364/365).

Line 370 add in "neutralized the omicron virus.."

Line 394: Reference needed

Line 413: This assertion seems to contradict what you said in the results (line 307/308)

Line 420: I am not sure you can say that correlated parameters indicate improved immune quality given these volunteers got a breakthrough infection, whereas those in vaccine only groups without correlations presumably didn't get any breakthrough infections over the same time period may actually have superior immunity as they were protected? You are implying that they will then have a subsequent improved immune quality based on other literature but you need to acknowledge that your findings could equally point to the opposite.

This discussion needs to include strengths and limitations discussion of the study, rather than repeating a lot of the findings. One e.g. of a limitation is alpha neutralisation was not assessed. Also it was an observational study not randomised etc.

Reporting summary:

Population characteristics: Please justify that participants were not discriminated based on demographics as this information not included in the paper

Recruitment: States "random employees".. does this mean volunteers were randomised? I don't think so as this was an observational study

Replication: Please clarify in the manuscript where technical replicates were performed and where not and justify/explain when they were not used

Data collection: Please include these dates in the manuscript

Flow cytometry: Please include all missing data on flow cytometry plots in figures (axis labels, axis scales, gate frequencies).

Reviewer #3 (Remarks to the Author):

This is an interesting and well-written manuscript defining the antibody and cellular immune responses to COVID-19 mRNA vaccination alone or in combination with breakthrough infections. Despite the low sample size in some groups (n=7), the authors find differences between immunization groups, showing more robust responses after omicron and delta breakthroughs. More specifically, the authors describe better antibody and B cell responses with 2 vaccine doses and delta breakthrough infection or three vaccine doses and omicron or delta infection than only 2 doses of vaccine with or without alpha breakthrough infection. Regarding T cell immunity, no differences were detected by the diverse immunization groups and overall T cell responses were low. A strength of the manuscript is that the variants of the breakthrough infections were identified by sequencing.

Main concerns:

-I disagree with the conclusion of the paper (line 427) that is also stated in the introduction (line70) regarding the 4th immunization not boosting the immune response except for the omicron breakthrough infection. The results do not support this statement as it is written and it may be misleading. In order to show a lack of boosting, the post-immunization response should be compared with the response just before the last immunization.

-Time between immunizations has an impact on the acquired immune response. This fact seems to be ignored in the manuscript. The alpha infections occurred very early after primary vaccination with the two doses. This is totally different from the groups having delta or omicron breakthroughs many months after primary infection. This should be acknowledged in the discussion as could explain why alpha breakthrough seems to induce poorer responses.

- There is no demographic and clinical information on the study participants and comparison between the study groups of the main demographic variables that have been associated with COVID-19 and immune responses. There may be confounders.

- Individuals were sampled 2-9 weeks following the last immunization. Were there differences between the immunization groups in the time since immunization? This could be a confounder too.

- How representative are the study individuals of the general population? Can the authors add this to the discussion?

-There are no limitations stated in the discussion.

-How was saliva collected? Sample collection may impact antibody measurements. Did all individuals have detectable antibodies? How is seropositivity in saliva determined? The analysis of the proportion of IgA/IgG would not make sense if done in groups that had almost no detectable antibodies. Can this be the reason for the higher proportion of IgA observed in the 2xVacc group?

- How are T cell responders (fig 4h) defined? Is any detectable T-cell response considered a positive response?

- The authors conclude that breakthrough infections induce better coordination of the immune response because responses are better correlated. However, is observing stronger correlations in plasma determinations really a reflection of better coordination of the acquired response? I would be more cautious with the statements related to better-coordinated responses.

- In the introduction, the authors refer many times to the protection of COVID-19 vaccines against infection, while current licensed COVID-19 vaccines are designed to protect against disease. It is well known that protection against infection and decrease of transmission by vaccination is poor. Booster doses are not needed to curb the SARS-CoV-2 transmission as it is mentioned in line 48, but to protect against severe disease. Also, the sentence in lines 57-58 does not make much sense because any infection would boost responses and the fact of having a breakthrough infection already means that responses were not optimal at that time.

Minor comments:

- Line 370: a couple of commas would increase clarity

**REVIEWER COMMENTS**

**Reviewer #1 (Remarks to the Author):**

In the manuscript “SARS-CoV-2 immunity following different combinations and breakthrough
infections” Pusnik et al. investigate the adaptive immune response after different combinations of
vaccinations (two vs. three doses) in the presence or absence of breakthrough infection. In addition,
they also investigate the immune response after breakthrough infection with different variants of
concerns (alpha, delta, and omicron). A scenario very much reflected in current real life. Importantly,
the variant of concern was determined by RT-PCR rather than inferring it from the time point of
infection. Overall, this is a very important and timely study as this answers the question if the times
of antigen exposure and/or the type of exposure (vaccination vs. natural infection) influences
adaptive immune responses.

**Minor comments:**

**Reviewer:** All samples were collected 2-9 weeks following the last immunization. Please clarify that
this time point also refers to breakthrough infections. i.e., 2 weeks after infection.... It would be
important to show that the average time point within these 2-9 weeks was comparable between
groups, as the adaptive immune response can change significantly within this time point.

**Response:** The reviewer raises an important point. With the term “immunization”, we wanted to
describe any type of challenge triggering an immune response including infection. However, we
agree that this might be confusing since immunization is often understood as a synonym for
vaccination. We now exchanged the term “immunization” for “antigen contact” and point out that it
covers both infection and vaccination throughout the article.

The individuals belonging to different immunization groups were preselected so that the times from
the last immune challenge, let it be infection or vaccination, did not significantly differ between the
groups (see graph below). We now added this information to the manuscript.

**Reviewer:** It would be interesting to compare the 3+Vacc+O group to a group that has been
vaccinated four times, as this would most accurately reflect the 4-time antigen exposure in both
groups.

**Response:** We agree with the reviewer, however, we were not able to collect samples of 4-times
vaccinated individuals during the study, since the 4th dose was not recommended for the general
population in Germany but rather for individuals at risk (e.g. elderly, pre-existing conditions).

**Reviewer:** Is it remarkable that previously vaccinated individual's to not develop T cell response
against the N protein. It would be good to include results from individuals that have only been
naturally infected and not vaccinated to demonstrate that N specific responses are detected in the
absence of vaccination.

**Response:** We agree with the reviewer and have added the point now to the manuscript. The
participants of the current study were all initially 2-times vaccinated. However, we previously
conducted studies where we measured N-specific CD4 T cell responses of individuals that were only
infected (<https://doi.org/10.1128/jvi.00760-22> figure 5a, and
<https://doi.org/10.1016/j.celrep.2021.109320> figure 2). Although the flow cytometry panels were
different, one can appreciate that only infected individuals had notably more IFN γ -expressing N-
specific CD4 T cells than the individuals from the current study that were initially vaccinated and
afterward acquired an infection. This is an important observation and was added to the discussion.
We thank the reviewer for his comments.

Reviewer #2 (Remarks to the Author):

The study by Pusnik et al presents a comprehensive immunogenicity combined vaccine with
breakthrough infection study. However, there are concerns below about cohort detail that
undermine the strength of the findings, as my main issue is that the timing of infection plays a bigger
role than the antigenic difference but is not sufficiently described- is the timepoint of sampling after
infection is comparable between groups (2-9 weeks?). The methods are sufficiently described. Some
statistical correlations are not adequately described. FACS gating strategies and gate frequencies are
not given.

Main comment:

**Reviewer:** Cohort details are needed for subject details as booster doses are recommended for at
risk groups which may differ from vaccine only groups.

**Response:** We agree with the reviewer and provide the cohort details as a supplemental table 1. The
third vaccination was recommended for the general population in Germany at the time of study and
most of the vaccinated individuals also received the third dose of vaccine regardless of their risk
status. Therefore, to our knowledge, there is no bias towards the high-risk population in any of the
groups. This information has now been added to the discussion. Moreover, none of the study
participants indicated to be immunodeficient. Two individuals in the 2xVacc group and one in the
3xVacc group were noted to have taken cortisol which acts immunosuppressive, however, their
immune responses were normal.

**Reviewer:** The timing between vaccination and infection is also likely different between the subjects
infected with different variants and therefore their boosting potential is either related to timing post
vaccination (i.e., infection at a short-term memory versus long term memory when neutralising
antibodies have waned is likely to recruit different immune parameters) and/or antigenic distance
and immune evasion from the ancestral vaccine. Furthermore, the sample timepoint after infection is
not given (e.g. N positivity rates after alpha infection- is this due to long sample timepoint?) and 2-9
81 weeks after infection samples were compared?

**Response:** The reviewer raises an important point. The timing between different immune challenges
let it be vaccination or infection is indeed different between the different groups since it reflects the
real-world situation where the onset of vaccination campaigns and prevalence of different SARS-CoV-
2 variants occurred at different times (see figure 1a). It was for example not possible to get samples
from vaccinated individuals that would acquire alpha and omicron infection at the same time after
vaccination since the omicron variant appeared much later than the alpha and most individuals got
vaccinated at the beginning of spring 2021. Therefore, we do not claim in the manuscript that we
compared the immunogenicity of different variants causing breakthrough infections, but rather
compare the immune response of individuals that underwent different immunization scenarios at a
given time after the last immunization. Together with antigenic distance, the severity of infection and
immunogenicity the time between immunizations contribute to the differences in immune response
observed for the groups assessed in the study. We added this important point to the discussion and
made it clear that we do not claim a comparison of immunogenicity against different variants.

The term immunization was used to describe any type of immune challenge including vaccination
and infection (We now exchanged the term “immunization” for “antigen contact” and point out that
it covers both infection and vaccination throughout the article), so 2-9 weeks apply to both
vaccination and infection as the last immunization. The individuals belonging to different
immunization groups were preselected so that the times from the last immune challenge, let it be
infection or vaccination, did not significantly differ between the groups (see graph below). We now
added this information to the manuscript.

**Reviewer:** Often immunisation is used to refer to vaccine or recovery from infection as a form of
immunization. This is confusing for the reader as immunisation typically means vaccine. If you want
to use the term this way throughout the article (including the way you refer to your different groups)
then this needs to be flagged in the introduction for clarity to the reader. The difference between
hybrid immunity (infection then vaccination) or breakthrough infection (vaccination then infection)
and immunization does not equate to vaccination or recovery from infection but vaccination
however is used interchangeably in the text.

**Response:** We thank the reviewer for pointing out this issue. The term immunization was used in the
manuscript to describe any immune challenge; either vaccination or infection. We now exchanged
the term “immunization” for “antigen contact” and point out that it covers both infection and
vaccination throughout the article.

**Reviewer:** No information on surveillance for infection method for this study. The commentary by
the authors in the results seem to suggest that all volunteers only had one infection i.e. alpha, delta
or omicron (i.e. lines 225-226) but what infection surveillance was carried out? Line 64: State here
what the study type was i.e. observational. There needs to be more information on how were
infections detected. And how often volunteers were surveyed. Was it just if volunteers developed

symptoms? Based on national surveillance? Through GP attendance? Individual/self-diagnosed
symptom detection?

**Response:** The volunteers with breakthrough infections were sampled and included in the study only
after the infection was confirmed by RT-PCR and the variant confirmed by sequencing as a part of
general surveillance at the University Hospital Bonn (as stated in the material methods section, study
cohort). As employees of the University of Bonn study participants were obliged to perform routine
self-testing twice weekly using antigen tests and were tested by RT-PCR whether they developed
symptoms. Moreover, individuals were excluded from the study if they had a history of previously
confirmed SARS-CoV-2 infection diagnosed elsewhere. All individuals with breakthrough infections
were infected once. For the groups without breakthrough infections, individuals without confirmed
SARS-CoV-2 infection, and negative nucleocapsid ELISA results were included. We have now added
this information to the study cohort description. We stated that the study is observational in line 64.

Minor comments:

**Response:** We truly appreciate the effort and would like to thank the reviewer for this in-depth
revision. We believe that these comments significantly improved our manuscript.

**Reviewer:** Abstract: line 27, were T cell responses impacted by infection?

**Response:** No. We changed the sentence to make it clear.

**Reviewer:** Introduction:

Line 40. introduction: Cut to just "However similar to infection-induced immunity..." and delete
"relatively" (line 41)- as relatively to what?

**Response:** We agree with the reviewer and accepted his suggestion.

**Reviewer:** Line 42. change to "By the time the first vaccines were rolled out/licenced/in use.." And
add reference for this sentence.

**Response:** We agree with the reviewer and accepted his suggestion.

**Reviewer:** Line 43. delete 'several'.

**Response:** We agree with the reviewer and accepted his suggestion.

**Reviewer:** Line 46. add official Pango lineage variant classification not just "delta" i.e. B.1.617.2 (also
for Alpha and Omicron infections).

**Response:** We agree with the reviewer and accepted his suggestion.

**Reviewer:** Line 47. Less frequent than what?

**Response:** Than among unvaccinated. We changed the sentence accordingly.

**Reviewer:** Line 48. Current vaccines do not significantly limit transmission, but boosters are
recommended to increase protection against severe disease.

**Response:** We agree with the reviewer and accepted his suggestion. We changed the sentence
accordingly.

**Reviewer:** Line 49, third and now fourth doses are offered in many developed countries. Update for
wider relevance.

**Response:** We agree with the reviewer and accepted his suggestion.

**Reviewer:** Line 51. add reference to support. Vaccines may have reduced vaccine efficacy against
Omicron compared to the ancestral virus, but still maintain high level of protection against mortality.

**Response:** We agree with the reviewer and accepted his suggestion.

**Reviewer:** Line 52- in which scenario does vaccine cause more severe disease? This is an anti-vaccine
sentiment that is not supported by a reference or data. Update or remove.

**Response:** We agree with the reviewer and changed the sentence.

**Reviewer:** Line 55/56 What is meant by 'equal immune response'? Antibody titre? T cell? More
specific parameters are needed.

**Response:** Neutralizing antibody titer, we changed the sentence accordingly.

**Reviewer:** Line 57: better protected from disease severity or infection risk? Be specific

**Response:** From severe disease, we changed the sentence accordingly.

**Reviewer:** Line 59 Change to " However, there is heterogeneity in breakthrough infections amongst
vaccinated individuals in terms of the number of previous immunizations and the variant that causes
the infection". Needs a reference.

**Response:** We changed the sentence accordingly.

**Reviewer:** Line 61 got -> became .

**Response:** We changed the sentence accordingly.

**Reviewer:** Line 64 Change to "amongst individuals" Change "the" to "their" or delete.

**Response:** We changed the sentence accordingly.

**Reviewer:** Line 65 add in alpha variant nomenclature

**Response:** We added it to line 39 where the alpha variant is first mentioned.

**Reviewer:** Line 65-69 I would suggest lower case “vacc” and capitalise Omicron etc throughout.

**Response:** Here we would prefer to keep V capitalized not to read it together with the x standing
before and keep the o (greek letter) for Omicron as all variants are labeled by greek letters.

**Reviewer:** Line 69: “In most cases?” What does this mean. Be specific i.e. was this significant or
proportion of subjects? Do you mean most immune end points or most volunteers or most of the
groups?

**Response:** We removed “In most cases”.

**Reviewer:** Line 71: This section should be moved to the discussion. What is meant by ‘more robust
immunity’? The magnitude of the response and at what timepoint following or during infection to
impact outcomes of infection for severity? The timing between vaccination and infection is also likely
different between the subjects infected with different variants and therefore their boosting potential
is either related to timing post vaccination i.e. infection at a short term memory versus long term
memory when neutralising antibodies have waned is likely to recruit different immune parameters)
or antigenic distance and immune evasion from the ancestral vaccine. From Fig 1A, though minimal
details on timepoint intervals are given.

**Response:** We removed this section from the introduction and will only discuss it in the discussion.
The higher magnitude of immune responses at the time of sampling was meant by “more robust
immunity”.

Please look at the response to major comment #2.

**Reviewer:** Methods:

Line 75: “Individuals”? – cohort details are needed for age/gender/timepoints/controls and date of
recruitment/study period.

What were the third and fourth vaccinations? Were these also mRNA vaccines? What determined if
they had third and fourth vaccines or didn’t? age groups or underlying conditions have been
prioritised for boosters- the cohort should be specified.

**Response:** We now provide the cohort details as a supplementary table 1. None of the study
participants received 4th vaccination. Also, 3rd vaccinations were mRNA vaccines (see supplemental
table 1). Third vaccinations were recommended for the general population.

Please also look at the response to major comment #1.

**Reviewer:** Line 81. “were not necessarily healthcare workers.” Is not sufficient cohort details.

**Response:** These individuals were employed at the University of Bonn or University Hospital Bonn.
Most of them might work in healthcare but not all, we did not ask participants to specify their
occupations.

**Reviewer:** Line 133: delete “the” add in version number of software (also needed in Reporting
summary).

**Response:** We changed the sentence accordingly and added the version number.

**Reviewer:** Line 150: pandemics -> pandemic.

**Response:** We changed the sentence accordingly

**Reviewer:** Line 158: change to “as a positive control, cells..”

**Response:** We changed the sentence accordingly

**Reviewer:** Line 170: change: “ acquired on a FACS Celesta”

**Response:** We changed the sentence accordingly

**Reviewer:** Results:

Demographic information needed included about the volunteers. Was there any underlying cohort
differences between the 7 groups? The timepoint intervals for sampling should be specified on Figure
1A, range, mean + stdev. Was this different between the variants? See main point above about
interval between infection and vaccination recruiting different immune responses.

What does Figure 1a represent? Did all volunteers get omicron infection at month 12?/ 3 months
after their 3rd vaccination?

**Response:** We now provide demographic information together with sampling dates and dates of
immunizations in supplemental table 1. Graphs representing the time between immunizations and
sampling were added as supplemental figure 1 due to the lack of space in Figure 1. Figure 1a is a
schematic representation of the immunization and sampling for different groups. It shows the
sequential order and approximate length of periods between the immunization and sampling events.
Detailed information is now provided as supplemental material.

Please also look at the response to major comment #2.

**Reviewer:** Fig1c and 1g for Omicron figures should have the same axis as the other two (scale 10-
10,000 vs 100-10,000)

**Response:** We believe tha this does not make sense since the neutralization of different variants is
compared in the next figure (1d). Y-axis ticks are automatically adjusted to visualize the data in the
best resolution (changing the axis to 100-10000 would compress the data points and would be
harder to distinguish; particularly bad for groups 2xVacc and 2xVacc+a).

**Reviewer:** Fig1d: Legend needs to state what the overlaid numbers refer to – IC50 medians?

**Response:** We added an explanation to the figure legend.

**Reviewer:** Line 183: revise title, “Not only the number but also the type of immunization is important
for potent humoral immunity against the SARS-CoV-2”

The study only describes mRNA immunization, in this case again the authors term immunization with
recovery from infection.

**Response:** The term immunization was used in the manuscript to describe any immune challenge;
either vaccination or infection. We now exchanged the term “immunization” for “antigen contact”
and point out that it covers both infection and vaccination throughout the article.

**Reviewer:** Line 185: And Ref, suggest Khoury Nat Med 2020.

**Response:** We thank the reviewer for the suggestion, the reference was added.

**Reviewer:** Line 224: The authors say the N antibody data complies with/reflects the S1 specific data
but where does it say what % volunteers seroconverted to S1-specific IgG? Fig1b shows significant
differences but not number/% volunteers. This information could be inferred by counting the dots
but it would be good to have this information specifically shown and/or the N antibody data
presented in the same way as Fig1b.

Also the authors state that the baseline is the 2xvacc but it would be good to see the data presented
against the pre-pandemic levels (WHO standard).

**Response:** Since the participants were all initially 2-times vaccinated they all had detectable anti-
Spike antibodies. This is now evident from supplemental table 1. For the anti-Nucleocapsid
antibodies, we only have information on whether the individuals were positive or negative since the
assay used is not quantitative.

The pre-pandemic levels are considered negative results since all used antibody determination assays
use measurements of plasma from individuals collected before the pandemic as a cutoff defining
seropositivity.

**Reviewer:** Line 253. Delete 'partially true', suggest: "In contrast to plasma, the only significant
correlations in saliva were ..."

**Response:** We changed the sentence accordingly

**Reviewer:** Fig2d: Y axis % of IgG -> should be Ig? How can IgG be less than 100% of itself? Are the
colours flipped or is this proportion of Ig? Again need legends for these figures: e.g. Fig2e what does
the shading indicate?

**Response:** There should be Ig instead of IgG, we corrected the mistake. The shading around the line
represents the 95% confidence interval around the line of best fit, we now improved the figure
legends.

**Reviewer:** Line 263: The FACS gating strategy and gate frequency should be added as supplementary,
especially for the definition of memory B cells (Fig 3a, 4a).

**Response:** The gating strategies have been added as supplemental figures 2 and 3.

**Reviewer:** Line 269/270: I don't agree with this conclusion is correct for IgM ie 2xvaccdelta looks very
similar to 2xvacc even with variability. I think best to state no statistical differences seen, though
there was a trend with some vaccine combinations such as IgA 2xvaccdelta, 3vacc and put in
discussion that this might be due to large intraindividual variability.

**Response:** We agree with the reviewer, and changed the sentence accordingly.

**Reviewer:** Line 277/278: Statistics needed for support.

**Response:** We changed the sentence and added a p-value.

**Reviewer:** Line 287: delete “the (change to “protection from severe disease”).

**Response:** We changed the sentence accordingly.

**Reviewer:** Figure 4: Further detail needed in legends- median and IQR shown?

**Response:** We now added further information to figure legends.

**Reviewer:** Line 297: need to be careful with wording given no significance. Say a trend not that they
were higher.

**Response:** The previous sentence states that no statistical differences were observed. The part
describing which groups were higher refers to the antibody data.

**Reviewer:** Line 300: Again can't say it was highest as not statistically significant. Also need to tell us
what the medians were.

**Response:** We make clear in the previous sentence that no statistical differences were observed. We
changed the sentence to make it clear. Medians are obvious from the boxplots that always show the
median.

**Reviewer:** Line 307-308: Is it meant that infection groups don't develop N-specific T cell responses?

**Response:** Yes, we made the sentence clear.

**Reviewer:** Line 312-318: Stats needed to support.

**Response:** There was a mistake in the graph 4g which has been corrected and the graph replaced.
We also changed the text and added p-values.

**Reviewer:** Line 319: Put in ref to fig4h after “functions”.

**Response:** The reference to the figure was added to the sentence.

**Reviewer:** Line 326-327: Again these conclusions need stats to back them up (from above
paragraph).

**Response:** We added the missing statistics.

**Reviewer:** Figure 5 annotation: Did you use both Pearson's and Spearman's as the figure legend
mentions both and the heatmap label is not titled with either.

**Response:** There was a mistake in the figure legend. We used Spearman's test.

**Reviewer:** Discussion:

Line 359: Given the timing of infection has not been disclosed, it is problematic to refer to B cell
memory and T cell memory responses based on a sampling timepoint as memory rather than

phenotype against Omicron/Delta etc. Unless the assertion is that these are cross-reactive vaccine
induced memory B cells based on the timepoint of sampling (which is not specified). Or if the cells
have been phenotyped for memory – please include gating strategy. Please clarify.

**Response:** We now provide the information regarding immunization and sampling time points as
supplemental material. Also, the gating strategy is provided as supplemental figures 2 and 3.

As all of these individuals have been immunized either with a vaccine or infection multiple times and
the sampling time point is minimum 2 weeks after the last immunization it is reasonable to assume
that the majority of detected B and T cells are memory cells. Moreover, the S-specific B cells that are
IgG+ or IgA+, are class-switched and therefore memory B cells. For the IgM+ S-specific B cells, CD27
was included to separate naïve from memory cells (see supplemental figure 2). For the T cells, no
additional memory marker was included in the panel and we agree with the reviewer that the
proportion of naïve or cross-reactive cells might be significant, especially in the case of N-specific T
cells. We, therefore, omitted “memory” when referring to the T cell responses.

**Reviewer:** Line 362: “responses”

**Response:** We changed the sentence accordingly.

**Reviewer:** Line 364: change to “of SARS-CoV-2”
Ref for this statement (line 364/365).

**Response:** We changed the sentence accordingly and added a reference.

**Reviewer:** Line 370 add in “neutralized the omicron virus..”

**Response:** We changed the sentence accordingly.

**Reviewer:** Line 394: Reference needed

**Response:** References were added.

**Reviewer:** Line 413: This assertion seems to contradict what you said in the results (line 307/308)

**Response:** Line 413 states that for some individuals N-specific T cells could be detected (probably
cross-reactive ones), which in our opinion does not contradict lines 307/308 saying that the groups
with breakthrough infection had levels of the N-specific T cells that were not significantly higher than
those of the only vaccinated groups.

**Reviewer:** Line 420: I am not sure you can say that correlated parameters indicate improved immune
quality given these volunteers got a breakthrough infection, whereas those in vaccine only groups
without correlations presumably didn't get any breakthrough infections over the same time period
may actually have superior immunity as they were protected? You are implying that they will then
have a subsequent improved immune quality based on other literature but you need to acknowledge
that your findings could equally point to the opposite.

**Response:** We agree that the fact that these individuals got infected might indicate that they
previously had inferior immunity (although exposure probably plays a more important role here),

however, we are describing the adaptive immune response at the time of sampling, so after the
immunity already got boosted by the infection.

**Reviewer:** This discussion needs to include strengths and limitations discussion of the study, rather
than repeating a lot of the findings. One e.g. of a limitation is alpha neutralisation was not assessed.
Also it was an observational study not randomised etc.

**Response:** We agree with the reviewer and now added limitations and strengths of the study at the
end of the discussion.

**Reviewer:** Reporting summary:

Population characteristics: Please justify that participants were not discriminated based on
demographics as this information not included in the paper

**Response:** Individuals were not discriminated by sex or gender when recruiting them for the study
and allocating them to groups, however, we collected information about sex and gender
(supplemental table 1).

**Reviewer:** Recruitment: States “random employees”.. does this mean volunteers were randomised? I
don’t think so as this was an observational study

**Response:** By “random employees” we wanted to say that they were not chosen based on any other
than criteria used for allocating them to the immunization groups. We agree that this might be
confusing, we removed “random” and made clear that the study is observational.

**Reviewer:** Replication: Please clarify in the manuscript where technical replicates were performed
and where not and justify/explain when they were not used

**Response:** We now specified in the methods section where technical replicates were performed.

**Reviewer:** Data collection: Please include these dates in the manuscript

**Response:** This information is now provided as supplemental table 1.

**Reviewer:** Flow cytometry: Please include all missing data on flow cytometry plots in figures (axis
labels, axis scales, gate frequencies).

**Response:** This information is now provided as supplemental figures 2 and 3.

**Reviewer #3 (Remarks to the Author):**

This is an interesting and well-written manuscript defining the antibody and cellular immune
responses to COVID-19 mRNA vaccination alone or in combination with breakthrough infections.
Despite the low sample size in some groups (n=7), the authors find differences between
immunization groups, showing more robust responses after omicron and delta breakthroughs. More
specifically, the authors describe better antibody and B cell responses with 2 vaccine doses and delta

breakthrough infection or three vaccine doses and omicron or delta infection than only 2 doses of
vaccine with or without alpha breakthrough infection. Regarding T cell immunity, no differences
were detected by the diverse immunization groups and overall T cell responses were low. A strength
of the manuscript is that the variants of the breakthrough infections were identified by sequencing.

**Main concerns:**

**Reviewer:** I disagree with the conclusion of the paper (line 427) that is also stated in the introduction
(line70) regarding the 4th immunization not boosting the immune response except for the omicron
breakthrough infection. The results do not support this statement as it is written and it may be
misleading. In order to show a lack of boosting, the post-immunization response should be compared
with the response just before the last immunization.

**Response:** We completely agree with the reviewer, our data does not show that the fourth
immunization is not boosting the response. With statements in lines 427 and 70, we wanted to
suggest that individuals with four immunizations do not have a higher immune response than those
with only three. We paraphrased the sentences accordingly.

**Reviewer:** Time between immunizations has an impact on the acquired immune response. This fact
seems to be ignored in the manuscript. The alpha infections occurred very early after primary
vaccination with the two doses. This is totally different from the groups having delta or omicron
breakthroughs many months after primary infection. This should be acknowledged in the discussion
as could explain why alpha breakthrough seems to induce poorer responses.

**Response:** The reviewer raises an important point that could help the interpretation of the results.
We now added this topic in the last paragraph of the discussion. It is difficult to say that the poorer
immune response after alpha breakthrough is solely due to the short time after vaccination. The
alpha breakthroughs were particularly mild, which could also be due to the high similarity between
the spike protein of the vaccine and the alpha strain.

**Reviewer:** There is no demographic and clinical information on the study participants and
comparison between the study groups of the main demographic variables that have been associated
with COVID-19 and immune responses. There may be confounders.

**Response:** We now added a table with demographic information that we collected about the
participants (Supplemental table 1). Different groups have similar age and gender constitutions as
now evident from supplemental figure 1.

**Reviewer:** Individuals were sampled 2-9 weeks following the last immunization. Were there
differences between the immunization groups in the time since immunization? This could be a
confounder too.

**Response:** There were no significant differences between the groups regarding time since the last
immunization. This is now evident from supplemental figure 1 and supplemental table 1.

**Reviewer:** How representative are the study individuals of the general population? Can the authors
add this to the discussion?

**Response:** Based on the demographic data the study cohort has gender and age distribution
resembling the general population (supplemental figure 1, supplemental table 1). The individuals
sampled were employed at the university at the time of specimen collection but were
nonhomogenous regarding education or work position (we did not collect this information). There
however might be bias towards highly educated individuals. We added this to the discussion.

**Reviewer:** There are no limitations stated in the discussion.

**Response:** We now added the limitations to the last paragraph of the discussion.

**Reviewer:** How was saliva collected? Sample collection may impact antibody measurements. Did all
individuals have detectable antibodies? How is seropositivity in saliva determined? The analysis of
the proportion of IgA/IgG would not make sense if done in groups that had almost no detectable
antibodies. Can this be the reason for the higher proportion of IgA observed in the 2xVacc group?

**Response:** Participants were instructed not to eat or drink at least 60min before the appointment.
During the sample, collection participants would retain saliva for 1-2 min and expectorate it in a 50
506 ml centrifuge tube. Saliva samples were then centrifuged at 12000g and 4°C to remove solid particles
and frozen. We have added this to the methods section.

Almost all individuals had detectable antibodies in saliva, the negative individuals were excluded
from the IgG/IgA ratio analysis. The positivity cutoff was defined by measuring the saliva of
individuals that have never been exposed to the virus and were also not vaccinated. We now added
this information to the methods section and Figure 2 legend.

**Reviewer:** How are T cell responders (fig 4h) defined? Is any detectable T-cell response considered a
positive response?

**Response:** All individuals that had a detectable T-cell response (higher than the DMSO stimulated
negative control) are considered as responders in fig. 4h. The information was added to the figure
legend.

**Reviewer:** The authors conclude that breakthrough infections induce better coordination of the
immune response because responses are better correlated. However, is observing stronger
correlations in plasma determinations really a reflection of better coordination of the acquired
response? I would be more cautious with the statements related to better-coordinated responses.

**Response:** Stronger correlations were not only observed between the immune parameters measured
in plasma but also between B and T cells. A higher degree of correlations within a group suggests that
the immunization scenario triggered (or suppressed if negative correlation) not only one immune
mechanism, like antibodies but also others like B and T cells; immune responses were coordinated.
Such multilayer immunity is generally considered stronger than for example only antibody-mediated
immune memory.

**Reviewer:** In the introduction, the authors refer many times to the protection of COVID-19 vaccines
against infection, while current licensed COVID-19 vaccines are designed to protect against disease. It
is well known that protection against infection and decrease of transmission by vaccination is poor.
Booster doses are not needed to curb the SARS-CoV-2 transmission as it is mentioned in line 48, but
to protect against severe disease. Also, the sentence in lines 57-58 does not make much sense

because any infection would boost responses, and the fact of having a breakthrough infection
already means that responses were not optimal at that time.

**Response:** We agree with the reviewer that the COVID-19 vaccines mainly protect from severe
disease and not infection. We, therefore, changed the manuscript text accordingly. We also changed
the sentence in lines 57-58 to be more precise.

Minor comments:

**Reviewer:** Line 370: a couple of commas would increase clarity

**Response:** We added the commas when necessary. Thank you.

REVIEWERS' COMMENTS

Reviewer #1 (Remarks to the Author):

The authors have addressed all concerns appropriately

Reviewer #2 (Remarks to the Author):

Pusnik et al, have thoroughly addressed all reviewer comments which have added clarity to the manuscript. There are however still minor issues as below.

1. The term 'antigen contact' is used in place of immunisation, but still does not seem appropriate. I would suggest 'exposure' or response instead. Supplemental figure 1 also needs updating on immunisation as a label- suggest exposure (fig d).
2. P and p are used, editorial comment will likely ask for consistency.
3. Methods comment: "As employees of the University of Bonn study participants were obliged to regularly perform antigen tests or RT-PCR whenever they developed symptoms similar to Covid-19." Was self testing only performed when symptoms occurred? ie would asymptomatic infections be detected in this study? (consider discussing this as a limitation – ie if self-testing was only done in the presence of symptoms we believe this is a limitation to this study not a strength).
Methods comment: "All participants were either employed or studied at the University of Bonn at the time of sampling but were not necessarily healthcare workers". Why would we think they were healthcare workers if they were working at the university? Or do you mean they were working/studying at a University Hospital? The discussion states "The study participants were employed at the University or University Hospital Bonn, therefore.." Were there two recruitment sites? Hospital and university?
Methods: Please provide further information about inclusion/exclusion criteria. You mention in the rebuttal that participants were selected based on time since vaccination or infection but were not selected on gender/age. Please make this clear in the manuscript. What other criteria do you look at if any? Also how were they recruited? By advertising? As this potentially impacts on assessment of bias/population cohort and hence it is important for the reader to be told.
4. Discussion comment: "Moreover, the frequency of these cells is considerably higher in only infected individuals suggesting that vaccination impairs the formation of nucleocapsid-specific T cells possibly by decreasing the severity of infection or due to the effect of original antigenic sin in vaccinated individuals.". This may be more that viral replication is reduced and therefore N-priming is reduced based on antigen availability rather original antigen sin which is a competitive response for N-related responses. I would reword here.
5. Discussion comment: "No biases towards high-risk populations were identified for any of the groups." Where in the results/supplementary is this presented? What high risk groups as. criteria did you look at?

Reviewer #3 (Remarks to the Author):

Thank you for addressing all comments and including additional information, I think the manuscript has been improved. However, I still have some comments:

- There is a statement that there are no significant differences in age and sex between the groups, but looking at the plots from Fig S1, it seems that there are substantial differences in medians and distributions. For instance, the median of for 2 vac or 3 vac groups was around 44-46 years, whereas for other groups such as 3 vac+ alpha is around 30 years. What statistical test was performed and were no differences overall or by pairwise comparisons?
- One of the limitations of the manuscript that is missing is the low sample size of the groups (n=7). Also, besides mentioning that the cohort might be biased towards highly educated individuals, it

should be added that it is probably not representative of the general population, also for other characteristics: they are healthy, relatively young, etc.

- Finally, despite some moderate and high correlations between plasma and cell determinations in groups with breakthrough infections, there is no evidence that it is the result of a better-coordinated response. Responses are associated but this does not mean that they are coordinated or less coordinated than in vaccination-only groups. You can discuss and speculate about better coordination in the discussion (I agree that "multilayer immunity is generally considered stronger than for example only antibody-mediated immune memory"), but for the figure title, description in results or in the abstract "correlations" should be used and not "coordination".

REVIEWERS' COMMENTS

Reviewer #1 (Remarks to the Author):

Reviewer: The authors have addressed all concerns appropriately

Response: We thank the reviewer.

Reviewer #2 (Remarks to the Author):

Pusnik et al, have thoroughly addressed all reviewer comments which have added clarity to the manuscript. There are however still minor issues as below.

Reviewer: The term 'antigen contact' is used in place of immunisation, but still does not seem appropriate. I would suggest 'exposure' or response instead. Supplemental figure 1 also needs updating on immunisation as a label- suggest exposure (fig d).

Response: We agree with the reviewer and changed “antigen contact” to “antigen exposure” as this better describes both immunization and infection antigen contact. We also updated the supplemental figure 1 as suggested by the reviewer.

Reviewer: P and p are used, editorial comment will likely ask for consistency.

Response: We thank the reviewer for bringing this up. We now use p throughout the manuscript.

Reviewer: Methods comment: "As employees of the University of Bonn study participants were obliged to regularly perform antigen tests or RT-PCR whenever they developed symptoms similar to Covid-19." Was self testing only performed when symptoms occurred? ie would asymptomatic infections be detected in this study? (consider discussing this as a limitation – ie if self-testing was only done in the presence of symptoms we believe this is a limitation to this study not a strength).

Response: Self-testing was mandatory and performed twice weekly also in absence of symptoms. RT-PCR was performed only for symptomatic cases. We now changed the sentence (line:356) to make this point more clear.

Reviewer: Methods comment: “All participants were either employed or studied at the University of Bonn at the time of sampling but were not necessarily healthcare workers”. Why would we think they were healthcare workers if they were working at the university? Or do you mean they were working/studying at a University Hospital? The discussion states “The study participants were employed at the University or University Hospital Bonn, therefore..” Were there two recruitment sites? Hospital and university?

Response: We agree that might be confusing, because some of the participants work at both sites the University and the University Hospital. There was only one recruitments site: the Hospital. We now made this clear in the manuscript.

Reviewer: Methods: Please provide further information about inclusion/exclusion criteria. You mention in the rebuttal that participants were selected based on time since vaccination or infection but were not selected on gender/age. Please make this clear in the manuscript. What other criteria do you look at if any? Also how were they recruited? By advertising? As this potentially impacts on assessment of bias/population cohort and hence it is important for the reader to be told.

Response: The individuals belonging to different study arms were preselected so that the times from the last antigen exposure did not significantly differ between the groups. Furthermore, individuals with a history of previous SARS-CoV-2 infection were not considered for this study. All individuals that had a breakthrough infection were infected only once. For the groups without breakthrough infections, only individuals without confirmed SARS-CoV-2 infection, and negative nucleocapsid ELISA results were included. There were no further selection/exclusion criteria. We now made clear in the manuscript that age and sex were not among the selection criteria. Individuals that fulfilled the selection criteria were recruited by the occupational healthcare department of University Hospital. The first contact was established by telephone after which a written invitation and a consent form was sent to the participants.

Reviewer: Discussion comment: "Moreover, the frequency of these cells is considerably higher in only infected individuals suggesting that vaccination impairs the formation of nucleocapsid-specific T cells possibly by decreasing the severity of infection or due to the effect of original antigenic sin in vaccinated individuals." This may be more that viral replication is reduced and therefore N-priming is reduced based on antigen availability rather original antigen sin which is a competitive response for N-related responses. I would reword here.

Response: We agree with the reviewer and changed the sentence accordingly.

Reviewer: Discussion comment: "No biases towards high-risk populations were identified for any of the groups." Where in the results/supplementary is this presented? What high risk groups as. criteria did you look at?

Response: We looked at the age of participants, since elderly are a high-risk group and we also questioned participants whether they take medications or have a condition that could compromise functioning of the immune system. As mentioned in the rebuttal: There were no significant differences in age distribution (supplemental figure 1). Two individuals in the 2xVacc group and one in the 3xVacc group were noted to have taken cortisol which acts immunosuppressive, however, their immune responses were normal. We now made this clear in the discussion.

Reviewer #3 (Remarks to the Author):

Thank you for addressing all comments and including additional information, I think the manuscript has been improved. However, I still have some comments:

Reviewer: There is a statement that there are no significant differences in age and sex between the groups, but looking at the plots from Fig S1, it seems that there are substantial differences in medians and distributions. For instance, the median of for 2 vac or 3 vac groups was around 44-46 years, whereas for other groups such as 3 vac+ alpha is around 30 years. What statistical test was performed and were no differences overall or by pairwise comparisons?

Response: We performed Mann-Whitney test with Holm's correction for multiple testing and no significant differences were observed. Also with pairwise t test and Holm's correction there are no significant differences between the groups. We agree that it might seem that the medians of some groups are different, however, this is due to the low number of samples in some groups and distribution of the data.

group1	group2	p (Mann test)	p (Mann test)	p (t test)	p (t test)
2xVacc	2xVacc+ α	1	ns	1	ns
2xVacc	2xVacc+ δ	0,169	ns	0,112	ns
2xVacc	3xVacc	1	ns	1	ns
2xVacc	3xVacc+o	1	ns	1	ns
2xVacc	3xVacc+ α	1	ns	1	ns
2xVacc	3xVacc+ δ	1	ns	1	ns
2xVacc+ α	2xVacc+ δ	1	ns	1	ns
2xVacc+ α	3xVacc	1	ns	1	ns
2xVacc+ α	3xVacc+o	1	ns	1	ns
2xVacc+ α	3xVacc+ α	1	ns	1	ns
2xVacc+ α	3xVacc+ δ	1	ns	1	ns
2xVacc+ δ	3xVacc	0,522	ns	0,226	ns
2xVacc+ δ	3xVacc+o	0,885	ns	1	ns
2xVacc+ δ	3xVacc+ α	1	ns	1	ns
2xVacc+ δ	3xVacc+ δ	1	ns	1	ns
3xVacc	3xVacc+o	1	ns	1	ns
3xVacc	3xVacc+ α	1	ns	1	ns
3xVacc	3xVacc+ δ	1	ns	1	ns
3xVacc+o	3xVacc+ α	1	ns	1	ns
3xVacc+o	3xVacc+ δ	1	ns	1	ns
3xVacc+ α	3xVacc+ δ	1	ns	1	ns

Reviewer: One of the limitations of the manuscript that is missing is the low sample size of the groups (n=7). Also, besides mentioning that the cohort might be biased towards highly educated individuals, it should be added that it is probably not representative of the general population, also for other characteristics: they are healthy, relatively young, etc.

Response: We agree with the reviewer and have now added low sample size as a limitation to the study. We also stated that the cohort is probably not representative of the general population due to the age and education (whether they are relatively healthy that we cannot say).

Reviewer: Finally, despite some moderate and high correlations between plasma and cell determinations in groups with breakthrough infections, there is no evidence that it is the result of a better-coordinated response. Responses are associated but this does not mean that they are coordinated or less coordinated than in vaccination-only groups. You can discuss and speculate about better coordination in the discussion (I agree that "multilayer immunity is generally considered stronger than for example only antibody-mediated immune memory"), but for the figure title, description in results or in the abstract "correlations" should be used and not "coordination".

Response: We agree with the reviewer and have now used "correlations" instead of "coordination" in the proposed sections.